



# One-to-one aeroservoelastic validation of operational loads and performance of a 2.8 MW wind turbine model in OpenFAST

Kenneth Brown[1], Pietro Bortolotti[2], Emmanuel Branlard[2], Mayank Chetan[2], Scott Dana[2],
Nathaniel deVelder[1], Paula Doubrawa[2], Nicholas Hamilton[2], Chris Ivanov[2], Jason Jonkman[2],
Christopher Kelley[1], and Daniel Zalkind[2]

[1]Sandia National Laboratories, Albuquerque, NM, USA
[2]National Renewable Energy Laboratory, Golden, CO, USA
**Correspondence:** Kenneth Brown (kbrown1@sandia.gov)

**Abstract.** This paper presents a validation study of the popular aeroservoelastic code suite OpenFAST leveraging weeks of measurements obtained during normal operation of a 2.8 MW land-based wind turbine. Measured wind conditions were used to generate one-to-one turbulent flow fields (i.e., comparing simulation to measurement in 10-minute increments, or bins) through unconstrained and constrained assimilation methods using the kinematic turbulence generators TurbSim and PyConTurb. A

total of 253 10-minute bins of normal turbine operation were selected for analysis, and a statistical comparison in terms of performance and loads is presented. We show that successful validation of the model is not strongly dependent on the type of inflow assimilation method used for mean quantities of interest, which have modeling errors generally within 5–10 % of the measurement. The type of inflow assimilation method does have a larger effect on the fatigue predictions for blade-root flapwise and tower-base fore-aft quantities, which surprisingly see larger errors from the assumed higher-fidelity assimilation

methods. Further work including improvements to the induction modeling in OpenFAST during high shear, as well as other possible improvements to the aerodynamic, blade, and controller modeling, may offer insight on the origin of the ∼5-40 % overprediction of fatigue for these quantities.

## 1   Introduction

Aeroservoelastic turbine models based on blade element momentum theory (BEMT) and equivalent beam models remain at the center of design and certification processes for wind turbines thanks to the balance they strike between accuracy and computational efficiency (Van Kuik et al., 2016). The multiphysics tool, OpenFAST (Jonkman and Sprague, 2021), which is actively developed at the National Renewable Energy Laboratory, is one of these models. Over the years, OpenFAST has been subject to several rounds of verification against other aeroservoelastic solvers (Rinker et al., 2020) and validation against measure-

ment (Guntur et al., 2017; Schepers et al., 2021; Asmuth et al., 2022; Boorsma et al., 2023), but changes in modern wind turbines, namely, the increased rotor size and concomitant changes in blade flexibility, blade aerodynamics, and atmospheric



forcing, suggest an ongoing need to validate OpenFAST at scales relevant to industry. Importantly, this validation should be accomplished with suitable assimilation of measured inflow into the simulation environment to obtain a synthetic wind field that matches as closely as possible the inflow experienced by the turbine.

So far, validation of OpenFAST relative to full-scale measurements has adopted either (1) non-turbulent and uniform inflow (Schepers et al., 2021; Boorsma et al., 2023), (2) purely stochastic turbulent inflow (i.e., based on the spectral magnitudes of a reference flow but with random phases) that matches time-averaged statistics of hub-height wind speed, hub-height turbulence intensity, and shear profile (Schepers et al., 2021), (3) time-resolved inflow at a single point in the domain (i.e., with more distant points reverting to random phases) that matches time-averaged statistics of hub-height wind speed, hub-height turbulence

intensity, and shear profile (Guntur et al., 2017), or (4) time-resolved inflow at multiple points in the domain (Asmuth et al., 2022). The strategy of combining one or more time series with a stochastic turbulence generation method as in (3) and (4) represents a compromise between the simpler approach of (2) that constrains the generated inflow only in terms of time-averaged statistics and emerging higher-fidelity approaches that combine large-eddy simulations with machine learning (Rybchuk et al., 2023). This paper adopts the second through fourth approaches, allowing comparison of the code predictions across different

levels of inflow assimilation methods.

    Recent efforts on other code suites have also compared across different levels of inflow assimilation methods. The data assimilation techniques considered by Pedersen et al. (2019) leveraged data from an upstream meteorological tower and included both unconstrained and constrained turbulence assimilation approaches. Surprisingly, the constrained turbulence approach increased the mean simulation errors by several percentage points for all damage equivalent loads (DELs) considered, and they

attributed this to possibly unmet assumptions about frozen turbulence and about the measured flow field passing completely through the rotor disk. However, the constrained approach did outperform the unconstrained approach when considering inflow data measured from a Pitot probe mounted on one of the blades. Nybø et al. (2021) used data from a meteorological tower as the input to a simulation study on the differences of tower-bottom fore-aft and blade-root flapwise DELs between unconstrained and constrained approaches. They found that the unconstrained approach produced 27 % and 12 % underprediction

of the tower-bottom fore-aft and blade-root flapwise DELs, respectively, compared to the constrained approach. Rather than using a meteorological tower or an on-blade sensor, Rinker (2022) constrained the turbulence fields to data generated from a turbine-mounted virtual lidar that sampled a simulated flow field. The constraining process produced a clear improvement of mean absolute errors for several quantities of interest (QoIs) versus unconstrained results. DEL predictions for the tower-base fore-aft bending moment improved when the constraint points from the lidar extended to at least 40 % of the rotor span from

the axis of rotation.

    Similar to the above three studies but considering instead the OpenFAST code suite, the objective of this effort is to assess the value of existing inflow assimilation tools of different levels of fidelity (i.e., including both unconstrained and constrained turbulence assimilation methods) for validation of simulations of Megawatt-scale wind turbine performance and loads. In addition to performing such a comparison across three levels of fidelity for the first time with OpenFAST, we here consider

other validation quantities (i.e., damage equivalent loads) not included in the previous, recent studies involving OpenFAST that employed one of the four inflow assimilation approaches listed previously (Guntur et al., 2017; Schepers et al., 2021; Asmuth



et al., 2022; Boorsma et al., 2023). In our work, quantitative comparison between measured and simulated turbine signals is calculated for 10-minute statistics of these QoIs:

- – Rotor speed

- – Blade pitch

- – Electrical power

- – Flapwise and edgewise blade-root bending moments

- – Fore-aft tower-base bending moment

The approach is an "end-to-end" validation, meaning that the accuracy of the inflow modeling, turbine aeroelastic modeling, and controller modeling are collectively evaluated according to the final turbine QoIs above. For the inflow modeling, we evaluate the relative merits of the several inflow assimilation methods with varying levels of simplifying assumptions as described above. For the turbine aeroelastic and controller modeling, a significant amount of attention was devoted to matching the behavior of the field turbine by inserting proprietary turbine and controller information into the sub-modules of OpenFAST.

The rest of the paper is organized as follows: The development of the turbine aeroservoelastic model is described in Sect. 2. Sect. 3 gives an overview of the methods for assimilation and modeling of the measured inflow data. Sect. 4 shows the results and is followed by suggestions for future experiments in Sect. 5. Conclusions are drawn in Sect. 6.

The study is part of the Rotor Aerodynamics, Aeroelastics, and Wake (RAAW) experiment, which is a collaboration between the National Renewable Energy Laboratory, Sandia National Laboratories, and the wind turbine manufacturer GE Vernova. The validation of the OpenFAST model presented here is based on measurements collected prior to the RAAW field campaign. The inflow and wind turbine measurements are therefore limited to instrumentation already present at the site before RAAW.

## 2 Turbine model development

The wind turbine used within the RAAW experiment is a highly-instrumented 2.8 MW prototype wind turbine that mounts a rotor of 127 m diameter at a hub height of 120 m. The turbine is located in Lubbock, Texas, in a region characterized by flat terrain. The first step of this study consisted of building the OpenFAST model of the turbine. This step was performed by combining different data sets shared by GE describing the aerodynamic and elastic properties of the rotor, the elastic properties of the rest of the turbine system, and high-level information about the controller. Additionally, the GE team shared experimental results from the structural testing of the blade and numerical results from its in-house solvers. All this information was used to develop an accurate OpenFAST model. GE's proprietary controller was replaced by the publicly available Reference OpenSource Controller (ROSCO) (Abbas et al., 2022), which was tuned to the reference information and turbine sensor data. The next sections elaborate on the process used to develop this model and on the verification and validation steps that were performed.





## 2.1 Aerodynamics

OpenFAST can simulate rotor aerodynamics at different levels of fidelity with models implemented in the module AeroDyn15. This study models rotor aerodynamics using BEMT. The blade aerodynamic shape and performance was discretized into 78 sections equally spaced along the blade span. A set of two-dimensional (2D) airfoil data blending clean and rough polars from wind tunnel measurements with a weighted average was shared by GE. The 2D polars were first interpolated adopting a piecewise cubic hermite interpolating polynomial scheme to match the spanwise distribution of thickness-to-chord ratio. Next, the polars of airfoils with thickness-to-chord ratios smaller than 0.7 were corrected to account for rotational effects adopting the Du–Selig model (Du and Selig, 1998). Note that the Du–Selig model relies on the successful identification of a linear regime of the lift curve, which is not obvious for airfoils located close to the blade root. This led to some arbitrary decisions about the range of angles of attack over which to apply the correction and to the decision of limiting the corrections to airfoils up to 0.7 of thickness-to-chord ratio. The airfoil unsteady behavior was modeled using an extension of the Beddoes–Leishman model developed by Minnema/Pierce (Damiani and Hayman, 2019). The parameters required by the unsteady airfoil aerodynamic model at each blade spanwise station were provided to OpenFAST as precomputed inputs.

A dynamic BEMT model that implements a continuous-time state-space form of Oye's dynamic model was used (Branlard et al., 2022). Also, the Glauert skewed wake correction model was used with flow expansion function of $15/32\pi$ (Pitt and Peters, 1980). Other corrections to the BEMT model included the Prandtl hub- and tip-loss models, wake swirl (tangential induction), and the influence of drag on the induction factors. Lastly, the effect of the tower on the incoming wind was accounted for using a potential flow model unless otherwise noted, and the tower aerodynamic loading was also included unless otherwise noted. The environmental conditions were tuned on a bin-by-bin basis using the data from the met tower (see Sect. 3).

## 2.2 Structural dynamics

The elastic response of the turbine was modeled by combining the reduced-order ElastoDyn beam model for the tower and the higher-fidelity BeamDyn model, which implements the geometrically exact beam theory (Jelenić and Crisfield, 1999; Wang et al., 2017) for the blades. The elastic properties for all turbine components were precomputed and shared by GE. The next subsections elaborate on the verification and validation steps that were performed to ensure the accuracy of the elastic model.

### 2.2.1 Elastic response of the blades

GE shared detailed elastic properties of the blades. An initial verification step was then performed to confirm that the data were converted to the BeamDyn coordinate system correctly. Figure 1 shows the static deflections and rotations for a single blade clamped at the root and subjected to gravitational loads in flapwise and edgewise directions. This verification step returned maximum offsets of 0.02 m in deflections and 0.5° in rotations and was considered satisfactory.

Next, a verification and validation step was performed by comparing numerical and experimental values of natural frequencies and values of structural damping for a single blade. The experimental values were generated during testing of the blade in a structural laboratory. Table 1 presents the percentage error of the blade natural frequencies and damping between the



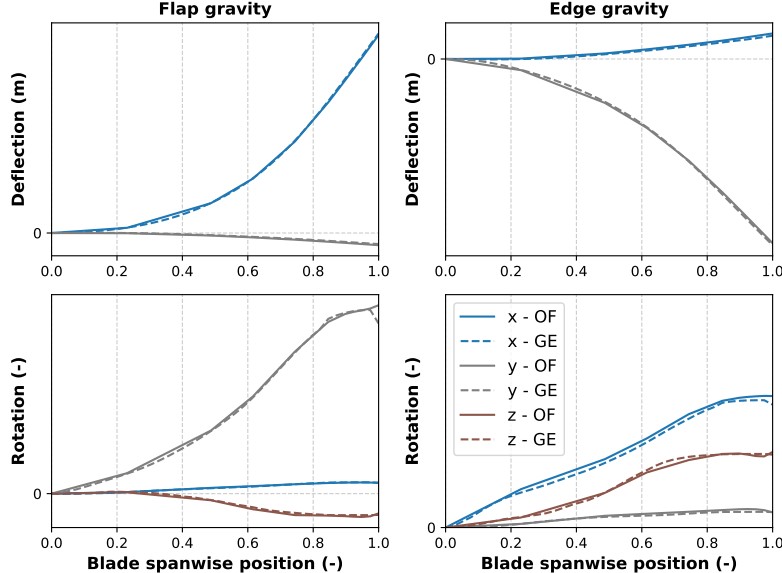

**Figure 1.** Deflection (top) and rotation (bottom) profiles of the blade loaded under gravity in the flapwise (left) and edgewise (right) directions. The dashed lines were generated by GE, and the solid lines were generated with BeamDyn, the beam model of OpenFAST implementing the geometrically exact beam theory. Results are expressed in the BeamDyn coordinate system, and rotations are expressed in terms of Wiener-Milenkovic parameters relative to the undeflected beam orientation.

laboratory test and the numerical predictions generated at GE and in OpenFAST. The match was again satisfactory, although some important discrepancies emerged, as described next.

The verification comparing the natural frequencies predicted by the two numerical models shows that the frequencies up to fourth flap, second edge, and first torsional modes match within 3 %, with negligible differences for the first flap, edge, and torsional modes. The validation step also shows a good match between the experimental results and the predictions of OpenFAST, with the exception of the first flap mode. Here, both models underpredict the natural frequency by 5.1 %. The research team could not explain the offset, which could have various origins, such as the impact of clamping of the blade root during laboratory testing.

The comparisons of structural damping show larger relative errors. Damping is an input to aeroelastic solvers and it is modeled differently across frameworks. BeamDyn models damping as a set of six stiffness-proportional values accounting for three rotations and three translations. This allows the user to set the desired damping for a mode of interest, usually the first or the second. In this study, three values of flapwise, edgewise, and torsional damping were initially set based on the experimental data from GE to match the first modes. These values achieved offsets of -5.9 % for the first flap mode and 1.3 % for the first edge mode. Because of the stiffness-proportional formulation, damping in the second modes was greatly overpredicted by OpenFAST compared to the results obtained in the laboratory. Despite this overprediction, the turbine in OpenFAST suffered from edgewise instabilities, which were resolved by artificially increasing the values of edgewise damping. The higher damping





**Table 1.** Comparison of blade natural frequencies and structural damping ratios between OpenFAST and reference data including experimental and numerical values shared by GE. The values provided are in terms of percent difference relative to the experimental ("Exp") and numerical ("Num") reference. Positive values represent higher frequencies and damping in OpenFAST relative to the values shared by GE.

|  | Natural Frequency | | Damping Ratio | |
|---|---|---|---|---|
|  | Exp (%) | Num (%) | Exp (%) | Num (%) |
| 1st flap | –5.1 | 0.0 | –5.9 | –5.3 |
| 1st edge | –1.1 | –0.1 | 1.3 | 16.2 |
| 2nd flap | 0.1 | 0.5 | 416.4 | 0.3 |
| 2nd edge | –1.2 | 3.0 | 192.7 | 29.0 |
| 3rd flap |  | 2.3 |  | 2.7 |
| 4th flap |  | 2.3 |  | –57.1 |
| 3rd edge |  | 11.3 |  | 341.6 |
| 5th flap |  | 12.7 |  | –7.9 |
| 1st torsion | 2.5 | –0.6 |  | –69.1 |

value is believed to account for additional sources of damping which may be present in the real turbine and currently lacking in OpenFAST, such as damping in the pitch system, main bearing, drivetrain, and yaw system.

### 2.2.2 Elastic response of tower and drivetrain

While the elastic response of blades was modeled at relatively high fidelity, the tower of the turbine was modeled in ElastoDyn, which implements a reduced-order beam model using the first two tower fore-aft and side-side modes. The fore-aft and side-side stiffness distributions and unit mass distribution were specified according to values provided by GE. The verification step consisted of verifying numerically the tower mass, which matched exactly, and then the natural frequencies with and without the rotor nacelle assembly. The results for the first four tower modes, namely first and second side-side and fore-aft, are reported in Table 2.

**Table 2.** Comparison of tower natural frequencies with and without rotor nacelle assembly (RNA) between OpenFAST and the numerical values shared by GE. The values provided are in terms of percent difference for the first four modes, namely first and second side-side and fore-aft tower modes. Positive values represent higher frequencies in OpenFAST relative to the numerical values shared by GE.

| Tower mode | No RNA (%) | RNA (%) |
|---|---|---|
| 1st side-side | -1.0 | +0.2 |
| 1st fore-aft | -1.0 | -1.1 |
| 2nd side-side | -4.0 | +1.0 |
| 2nd fore-aft | -4.0 | +2.5 |





The comparison returned small discrepancies between natural frequencies by OpenFAST and by the numerical solver at GE.

The discrepancies were attributed to the different model fidelity and to different discretization of the tower properties along the its height. In terms of damping, the values for the tower structure were assigned to the individual modes. Finally, the stiffness and damping of the drivetrain system were populated thanks to data shared by GE.

## 2.3 Aeroelastic response of the turbine system

Next, the aeroelastic modeling of the full turbine was verified and validated during turbine operation.

### 150 2.3.1 Experimental modal analysis

The experimental natural frequencies were obtained by calculating the power spectral density (PSD) for signals from strain gauges installed at the blade root and tower base. In signal processing, there are several ways of converting a signal from the time domain to the frequency domain. Choosing the correct method depends on the data or signals in question. In this scenario, Welch's averaged, modified periodogram method with a Hanning window was used to convert time series data to the frequency

domain. This method was preferred, as its approach to periodogram estimations helps reduce noise in the power spectra.

The analysis of the experimental data used for modal analysis was split into two sections: emergency stop and normal operation. Data from emergency stops were helpful for finding component natural frequencies without the influence of wind speed. The normal operation data were binned by rotor speed between cut-in and rated.

Conducting the experimental modal analysis with the existing set of installed sensors came with several challenges. A critical

limitation was that the only measurement location along the blade span was the strain gauges close to the root. Therefore, it was not possible to extract any information regarding modal shapes. Fundamentally, the gauge measurements only allowed us to derive the PSDs of the blade root strain.

When analyzing the PSD, it was found to be difficult to isolate and find the component frequencies. Consequently, the peaks were saturated and extremely difficult to identify experimentally, especially when we chose to run a blind comparison to the

165 numerical model. It was particularly difficult to find the first and second blade-root flap frequencies during normal operation. We were only able to extract the first flap frequency from emergency stop data. Additionally, there was high uncertainty related to the second blade root edge frequency because the peaks varied between data files and there were instances where there was no energy in the expected region. We tried to apply a method known as time synchronous averaging, which can help remove the rotor passing frequencies; however, this would have required a much higher data sampling frequency to be successful.

### 170 2.3.2 Validation of modal analysis

The relative difference between experimental frequencies extracted using the strain gauge data and the numerical frequencies estimated using OpenFAST is shown in Fig. 2 across a range of rotor speeds. For the first flapwise modes, we observe a good agreement at lower rotor speeds and growing discrepancies at higher speeds. Note that this trend is opposite to the one reported in Table 1, where OpenFAST underpredicts the natural frequency of the first flapwise structural mode. The different trend is




attributed to limitations of the linearized unsteady aerodynamic model and to the uncertainty in the experimental measurements caused by the high aerodynamic damping, which makes experimental frequencies hard to identify. The numerical and experimental first and second edgewise modes match better, with differences within 8 %. The better match is explained by the small impact that rotor aerodynamics have on edgewise modes and by a more precise determination of the experimental frequencies of the system. Note that edgewise modes are more important than flapwise modes because they are usually affected by low

damping and are prone to aeroelastic instabilities (Volk et al., 2020). Lastly, tower modes match within 2 % at low rotor speeds, growing to ±15 % toward rated rotor speed. Again, the impact of aerodynamics is thought to be responsible for the growing offset.

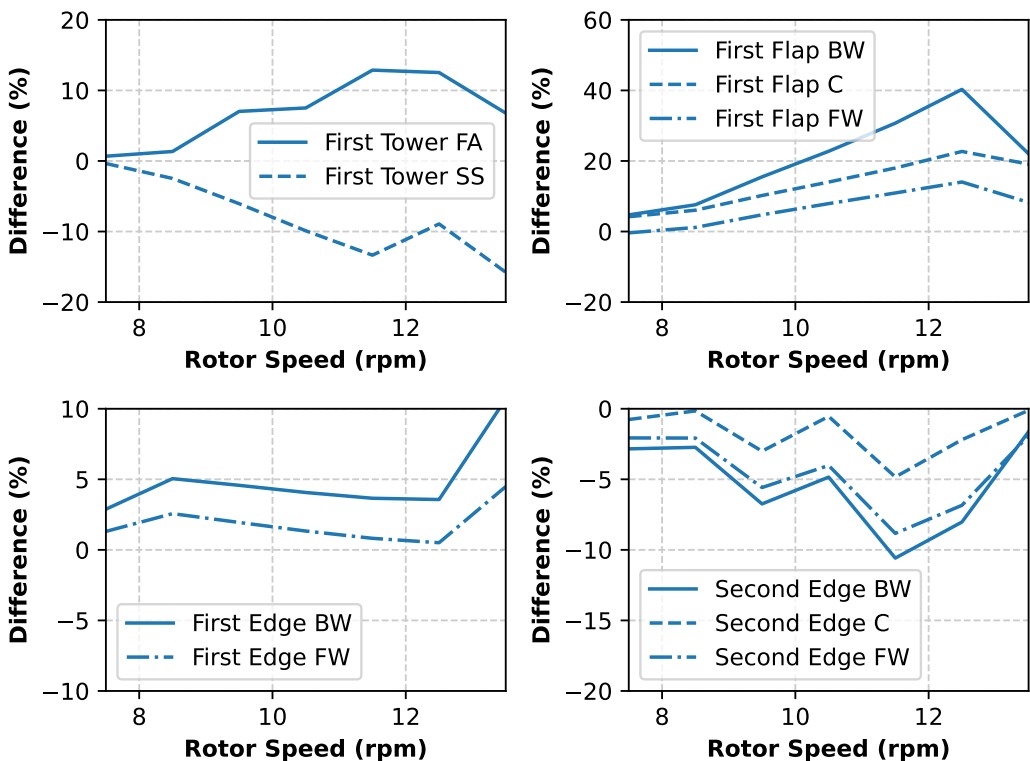

**Figure 2.** Relative differences between numerical (OpenFAST) and experimental (from blade-root strain gauges) natural frequencies for first tower modes (top left), first rotor flap modes (top right), first rotor edge modes (bottom left), and second rotor edge modes (bottom right). For the tower, fore-aft (FA) and side-side (SS) modes are reported. For the rotor, backward whirling (BW), collective (C), and forward whirling (FW) modes are reported. Positive values represent higher frequencies in OpenFAST relative to experimental values.





## 2.4 Controls

While the ideal validation process would incorporate the actual turbine controller in the turbine model, such as in Zierath
et al. (2016), this was not possible due to concerns around intellectual property. The solution to this problem was to adopt the
ROSCO controller (Abbas et al., 2022), which was coupled to OpenFAST to match the steady-state and transient behavior of
the field controller observed through historical SCADA data. The ROSCO generator speed set points were used to match those
of the field controller. The peak shaving, or thrust limiting, parameter of ROSCO was used to reproduce the mean and peak
blade and tower loading near rated power; this also resulted in similar near-rated power production. Step wind simulations were
used to tune the transient response of the torque and pitch controller bandwidths to match the GE controller's response to the
same wind input. Because of the long run times for OpenFAST simulations with BeamDyn, we ran 72 step wind simulations
in parallel with uniformly distributed ROSCO tuning inputs (pitch and torque control bandwidths). We then evaluated the
difference in the generator speed and rotor thrust response between the GE reference and our controlled OpenFAST model.
The simulation parameters with the lowest error were used to prescribe the parameters for another set of 72 simulations in a
smaller design space. From these simulations, the best combination of low generator speed and rotor thrust error were used to
select the set of ROSCO tuning parameters. Since both the generator speed and rotor thrust error could not be simultaneously
optimized, some judgment was used to give slightly more weight to the generator speed response error as it is a more direct
measure of the controller's desired behavior.

Although available in ROSCO, individual pitch control (IPC) and tower damping control features were not enabled because
the controller logic used to activate these features in the proprietary field controller was unknown. Even so, a generally good
agreement between the field controller and ROSCO was realized, as will be demonstrated in Sect. 4.1.

SCADA data shows more sophisticated system monitoring and control mode switching in the GE controller, compared to
ROSCO. These differences in the transient behavior in low- and near-rated wind speeds could lead to small discrepancies in
fatigue load results.

## 3 Inflow assimilation methods

This section describes how field data were used to generate one-to-one inflow bins for the numerical simulations.

### 3.1 Experimental campaign

The validation of the OpenFAST model presented here is based on measurements collected prior to the RAAW field campaign.
The inflow and turbine measurements are therefore limited to instrumentation already present at the site before RAAW.

In terms of inflow, we focused on assimilating data from the meteorological tower, which is shown in planform view in
Fig. 3 along with the turbine. The meteorological tower was instrumented with various wind sensors, including three RM
Young 81000 ultrasonic anemometers, five Thies Clima First Class cup anemometers, three MetOne 020WD wind vanes, and
a ground-sitting WindCube lidar, the latter of which provided only 10-minute statistics during the period considered here.





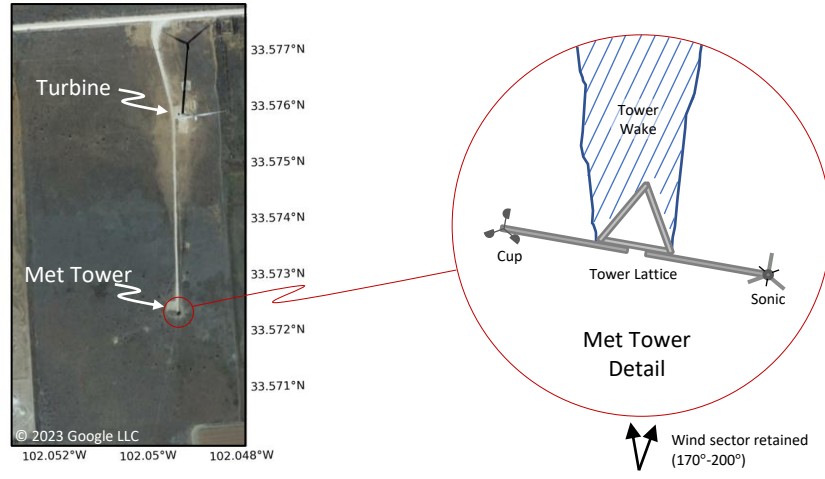

**Figure 3.** Planform view of the test site, including inset showing the cup and sonic anemometers in relation to the wake of the tower lattice for the wind sector retained in this study.

In this study we preferred to use data from the ultrasonic anemometers rather than data from the cup anemometers with co-elevated wind vanes because of the inclusion of the $w$-component of velocity in the ultrasonic measurement and because of a malfunction in the top-tip wind vane during the campaign. The 10-minute WindCube data were used to remove static wind direction offsets in the ultrasonic data that ranged between -9° and 14° between the various anemometers, likely due to misalignment during installation of the ultrasonics with the cardinal directions. The three ultrasonic anemometers spanned nearly the full height of the rotor, mounted onto booms at 52.6 m, 110.5 m, and 179.5 m. The historical data analyzed herein were output at a frequency of 1 Hz (note, this value was later raised to 20 Hz for the continuation of the data collection happening within the RAAW experiment).

Inline with the objectives presented in Sect. 1, the wind turbine channels of interest for this study included the rotor speed, blade pitch, electrical power, and the blade-root and tower-base bending moments, the latter two of which were sensed with strain gauges located near the blade roots and tower base, respectively. Re-calibrations of the strain gauges were performed such that the calibrations were never out of date by more than 90 days. Even so, we estimate an uncertainty of up to 150 kN-m in these measurements based on changes in some of the calibrations over such intervals.

The validation data in this article were collected between 22 September 2021 and 14 May 2022, and data are organized into 10-minute bins during this period. For each bin, several preprocessing steps were applied to the ultrasonic data to render the data appropriate for model validation. First, following Kelley and Ennis (2016), who processed 2.5 years of meteorological tower ultrasonic data from the nearby 200 m tower run by Texas Tech University, several quality control filters were applied to the $u$, $v$, and $w$ signals:

– remove all values above an absolute magnitude of 30 m s$^{-1}$





- remove all values that are identically zero

- remove all remaining values that are deemed spikes, or statistical outliers in the time series data, as identified using a
median absolute deviation filter with a time window of 300 s and a threshold of $5 \times 1.4826 = 7.4130$

An additional filter used in Kelley and Ennis (2016) to remove values based on a hold detection criterion was omitted from the current work since this filter appeared to be eliminating valid data in some cases.

Next, bins that had less than 95 % remaining data availability or had time spans of removed data longer than 5 s were also rejected. For the accepted bins, any instances of scattered data removal were filled in with a cubic hermite interpolation.
Summary statistics were then calculated over 10-minute bins. Finally, the horizontal components of the velocity series at each ultrasonic height were rotated into the reference frame of the 10-minute-mean reading of the hub-height wind vane, which was the appropriate form for input into the inflow assimilation methods.

In addition to the preprocessing steps above, the wind and turbine data of each 10-minute bin also underwent several filtering processes to be deemed valid and useful for the validation analyses. These filters included bounds to eliminate bins that in-
cluded malfunctioning sensors, bins with an idling turbine, bins including turbine start-up/shutdown events, bins with absolute mean yaw misalignment greater than $10°$, bins with yaw standard deviation greater than $4°$, bins with absolute mean shear exponent greater than 1, bins with absolute mean veer (as linearly fit from measurements taken from 52.6 m to 179.5 m above ground level) greater than $50°$, and lastly bins where the wind direction deviated more than $15°$ from the $185°$ heading of the meteorological tower relative to the turbine. This last condition not only prevented the ultrasonic sensors from ever being
waked by the mounting boom arms, the meteorological tower structure, or the wind turbine, but it also meant that specific turbulence structures passing through the rotor disk were more likely to be the same as those sensed by the meteorological tower, assuming a frozen turbulence hypothesis.

The above filtering reduced the data set to 253 10-minute bins, or 1.8 days, for validation analysis. Figure 4 displays 10-minute inflow statistics from these bins, which demonstrate a diversity of inflow conditions and thus imply a relatively broad
range of turbine operating conditions. The lack of cases with turbulence intensity above 0.15 in Fig. 4(a) is a consequence of the filtering on yaw standard deviation as described above, which was required because of limitations in the modeling setup. It is also noted that the existence of some relatively high shear exponents in Fig. 4(b) is a known characteristic of the site, and cases with these conditions were retained in the data set though some validation error should be expected for such cases since IPC was not included in our turbine model and was active in the field turbine in these cases. We elected not to filter such
cases from the data set because many 10-minute bins had some IPC activity but few bins had persistent IPC activity. Data for rotor-height veer from the wind vanes is omitted from Fig. 4(c) because of non-physical wind direction shifts that were observed in the readings of the wind vane near the top-tip position. However, the wind vane near hub height, which was used to rotate the ultrasonic data into the appropriate reference frame as described previously, was reliable.





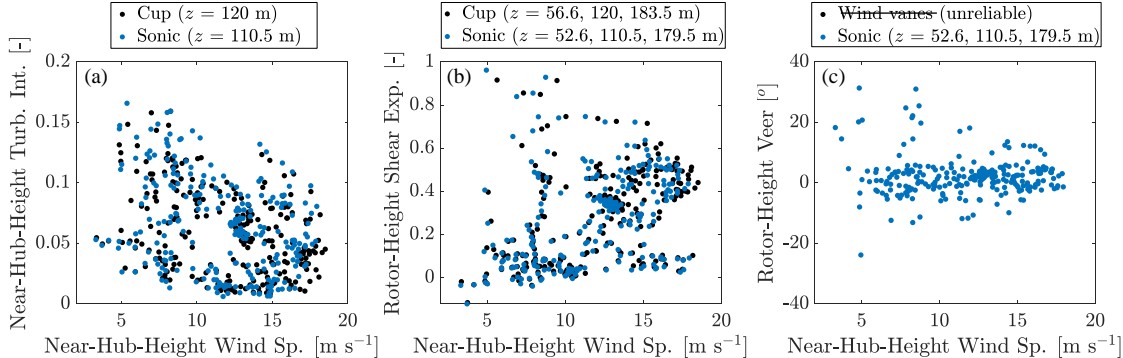

**Figure 4.** Mean inflow conditions for the 253 10-minute bins selected for model validation: (a) horizontal turbulence intensity near hub height, (b) shear exponent calculated from a power-law fit over the rotor span, and (c) veer calculated from a linear fit over the rotor span. The data are plotted versus horizontal wind speed as measured by the ultrasonic anemometer near hub height (at 110.5 m).

## 3.2 Computational environment

The inflow simulations were performed in three different ways by using two different sets of inputs to TurbSim (Jonkman, 2014) as well as an implementation of PyConTurb (Rinker, 2018). These kinematic turbulence generators begin with information on the spectra and spatial coherence of velocity components, which are then translated to the time domain via an inverse Fourier transform. Each generator can produce Gaussian turbulence fields according to a spectral model, which is herein taken as the Kaimal spectrum with exponential coherence as defined in the International Electrotechnical Commission's (2005) 61400-1 standard.

The differences among the three approaches, which are described in Table 3, center around how the measured data are assimilated. The turbulent fluctuations in the baseline TurbSim approach, termed here TurbSim simple, are stochastic and based on the turbulence intensity measured by the ultrasonic anemometer at the near-hub-height location. The Fourier magnitudes are determined from the spectral model, which is scaled so that the turbulence intensity matches that which was specified. Random phases are applied to the Fourier terms, and the phases of the streamwise component of velocity are correlated based on the spatial coherence model. The two higher-fidelity approaches of generating inflow are TurbSim with the TIMESR option and PyConTurb. These two approaches constrain the turbulent time series to match time-resolved measurements by linearly interpolating Fourier magnitudes from the measured time series to the computational grid and constraining the Fourier phases to match the wind series provided at one or more measurement locations. In TurbSim TIMESR, only one point in the domain can be constrained. PyConTurb, in contrast, is able to apply the same constraints to an arbitrary number of points. For the simulations performed here, TurbSim TIMESR is constrained based on the near-hub-height ultrasonic anemometer measurements. In PyConTurb, all three ultrasonic anemometer measurement locations are used. For each turbulence assimilation method and each 10-minute bin, six turbulence seeds were generated to improve statistical convergence over the non-constrained data





of the turbulence grids. As a result, 4554 turbulent inflows were produced (253 10-minute bins, 6 random seeds, 3 different approaches).

Other differences between the three methods are related to how the mean wind speed and direction are assimilated from the measurement. As shown in Table 3, TurbSim simple has the most restrictive assumptions in this regard, as it only generates power-law wind profiles without veer. The two other methods are more flexible, though the linear interpolation of wind data

for these methods is likely to result in wind speed and direction profiles that do not exactly match the observed conditions, especially in stably-stratified conditions.

For all three methods, the turbulence plane was a lateral-vertical grid of 33 by 33 points that was 10 % wider than the rotor diameter both laterally and vertically (see Fig. 5). The TurbSim methods include tower nodes to simulate wind loading on the entire tower (Jonkman, 2014), whereas PyConTurb does not have this feature. Tower aerodynamics were therefore disabled for

the PyConTurb cases.

**Table 3.** Comparison of inflow assimilation methods. Single-point constraints are enforced from the ultrasonic anemometer near hub height (i.e., z = 110.5 m), and three-point constraints are enforced from all three ultrasonic anemometers.

| | TurbSim Simple | TurbSim TIMESR | PyConTurb |
|---|---|---|---|
| Turbulence Method | Unconstrained Kaimal (Turb-Model=IECKAI) | Constrained Kaimal at single point with exponential coherence (TurbModel=TIMESR) | Constrained Kaimal at three points with exponential coherence |
| Turbulence Magnitudes | Uniform (derived from single point[1]) | Linear interpolation from three-point input | Linear interpolation from three-point input |
| Spatial Coherence | IEC in $u$-component, none currently enforced in $v$ and $w$ | GENERAL[2] in $u$-component, none currently enforced in $v$ and $w$ | IEC in $u$-component, none currently enforced in $v$ and $w$ |
| Wind Speed Profile | Power-law interpolation from three-point input | Linear interpolation from three-point input | Linear interpolation from three-point input |
| Wind Direction Profile | None enforced | Linear interpolation from three-point input | Linear interpolation from three-point input |

Figures 6, 7, and 8 show an example of the simulated inflow versus the measurement for a sample 10-minute bin captured on 16 March 2022. The data here illustrate that TurbSim simple generally matches only the 10-minute statistics while the constrained turbulence assimilation methods (i.e., TurbSim TIMESR and PyConTurb) additionally match the time-varying values at one or more measured heights. The small differences in the data of TurbSim TIMESR and PyConTurb compared to

the measurement at the constraint points are due to non-alignment of the simulation grid with the measurement location, and PyConTurb notably has two more constraint points than TurbSim TIMESR, as illustrated by matching of the measurement data

---

[1]The near-hub-height velocity time series is linearly detrended before calculating turbulence intensity as per Larsen and Hansen (2014), and ScaleIEC is set to 1 to enforce the exact value specified near hub height given the desired sample rate

[2]See Jonkman (2014) for description of the coherence model



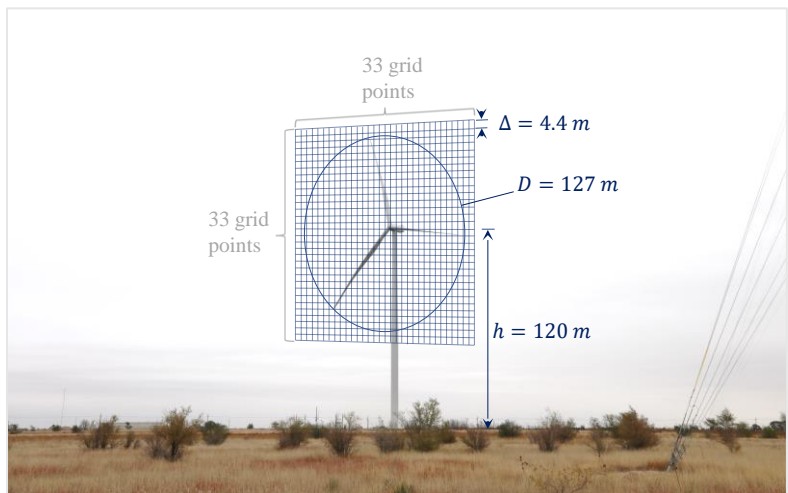

**Figure 5.** Image of the 2.8-MW GE wind turbine in Lubbock, Texas, U.S.A. with turbulence grid overlaid.

by PyConTurb at the top-tip and bottom-tip data in each figure. Note that the detrending process for TurbSim simple described in the footnotes of Table 3 results in significantly lower turbulence levels for TurbSim simple in this (and some other) example bins because of the time-gradient in wind speed during this interval. The corresponding difference in turbulent energy does not

manifest in the spectral plots of Fig. 7 because the data have been binned on 60-second intervals, thus eliminating the long-pass time scale that is affected by the detrending process.

For reference, the 10-minute statistics of another sample bin are given in Fig. 9. This bin demonstrates a non-monotonic shear profile (i.e., jet) that is sometimes observed at the site. TurbSim simple is not able to capture the shape of the shear profile, whereas the two higher-fidelity approaches can roughly capture the shape within the limitations of linear interpolation.

A note is appropriate about the frequency of the wind field input to OpenFAST, which is unaltered from the 1 Hz sampling frequency of the meteorological tower described previously. For calculations involving DEL, neglecting frequency content in the wind higher than 1 Hz will lead to slight underprediction of DELs. For a turbine of similar size and rated rpm, Sim et al. (2012) demonstrate an underprediction of 5 % and 2 % for the blade-root flapwise DEL and tower-base fore-aft DEL, respectively, by using a 1 Hz inflow and a coarse turbulence grid compared to an inflow with higher temporal and spatial

resolution. In this study, we make no attempt to populate the higher-frequency content of the measured inflow but note that the measured 1 Hz corresponds to more than 4 times the rotor revolution frequency (>4P) at rated rpm, which allows excitation of the 3P frequency. Frequencies of 6P and 9P, which might especially affect the tower, are not present in the simulation at rated rotor speed.

It also should be considered that no temporal offset was applied to the meteorological tower readings to account for the

advection time of the flow between the meteorological tower and the rotor, and this time is on the order of 20–100 s depending





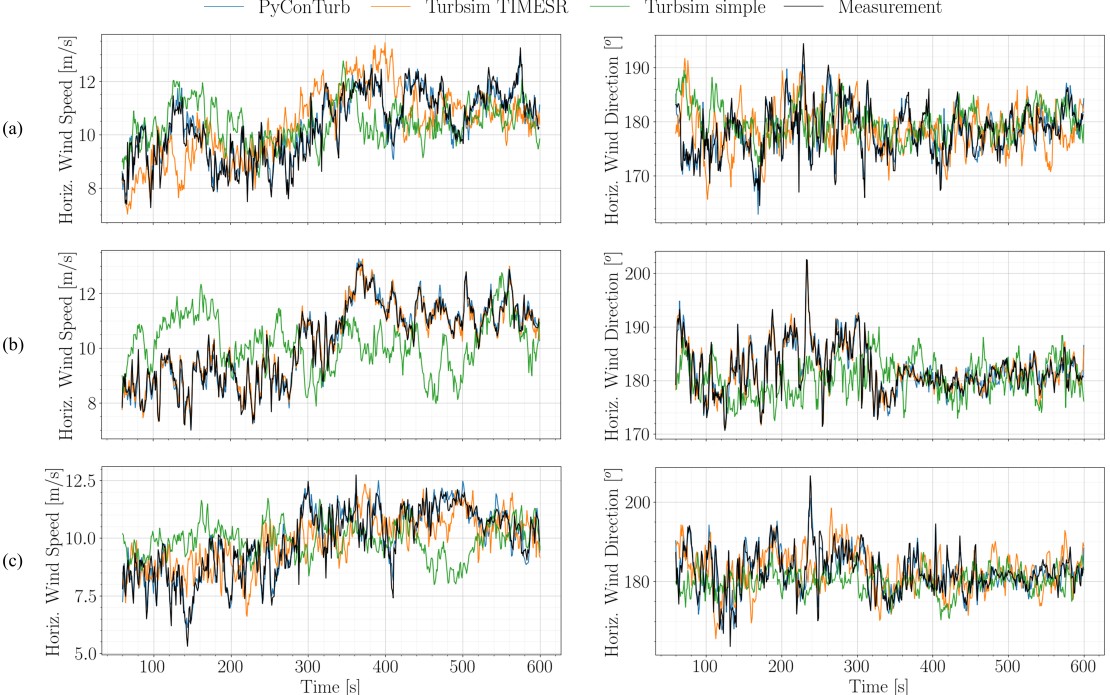

**Figure 6.** Comparison of time series of simulated inflow to measured inflow for a sample 10-minute bin for the sonic anemometers near (a) top tip, (b) hub height, and (c) bottom tip. The simulated data have been interpolated from the computational grid to the exact measurement locations, which were 179.5 m, 110.5 m, and 52.6 m, respectively. Only one of six turbulence seeds is shown.

on the wind speed. In the comparison of a large number (i.e., 253) of 10-minute bins drawn from all times of day, we expect this time offset to add noise but not bias to the comparisons.

## 3.3 Postprocessing

Outputs from the field turbine and simulated turbine were processed identically. The first minute of each 10-minute bin was
discarded to remove transients in the simulated results. Bending moment data from the simulations were interpolated to the position of the strain gauges at the blade roots and tower base. Bin averages were calculated, and DELs were calculated as in the OpenFAST Python toolbox (Branlard et al., 2023) at a frequency of 50 Hz, which was the rate of the measurement and simulation output.

## 4 Results

This section describes the results of the one-to-one validation beginning with analysis of the basic operability (i.e., mean rotor speed and blade pitch) and proceeding to power, blade loading, and tower loading.

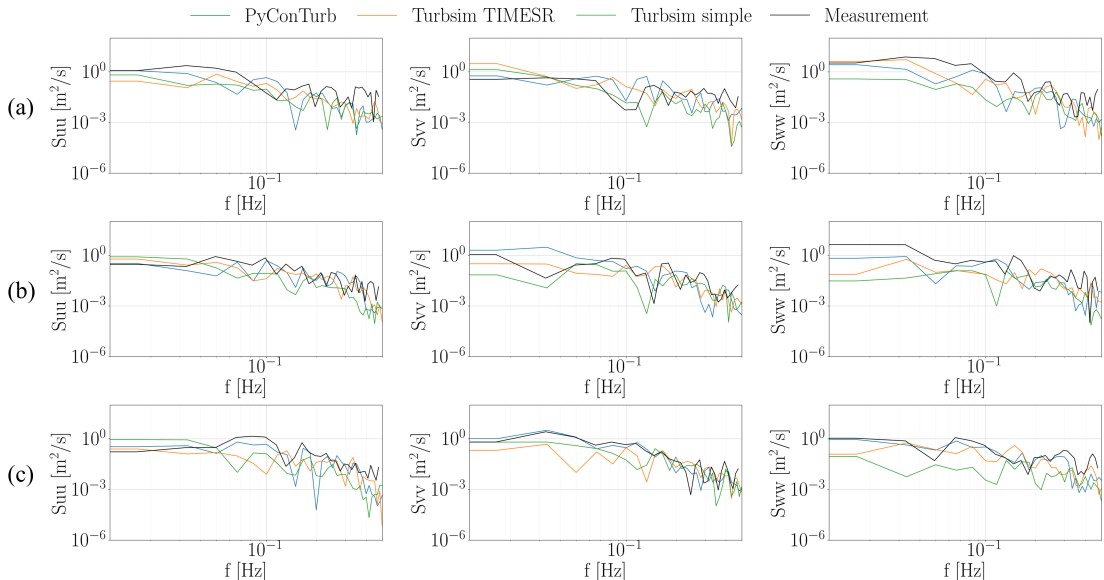

**Figure 7.** Comparison of turbulence spectra of simulated inflow to measured inflow for a sample 10-minute bin for the sonic anemometers near (a) top tip, (b) hub height, and (c) bottom tip. Before calculating the spectra, the simulated data have been interpolated from the computational grid to the exact measurement locations, which are 179.5 m, 110.5 m, and 52.6 m, respectively. Spectra are calculated using 60 s bins and a Hanning window. Only one of six turbulence seeds is shown.

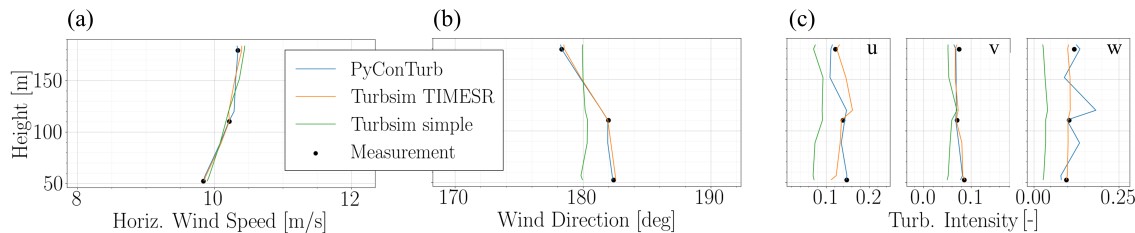

**Figure 8.** Comparison of vertical profiles of simulated inflow to measured inflow for a sample 10-minute bin for (a) horizontal wind speed and (b) direction, and (c) the three components of turbulence intensity, which are calculated as the standard deviation of the given component divided by the mean of the $u$ component. Only one of six turbulence seeds is shown.



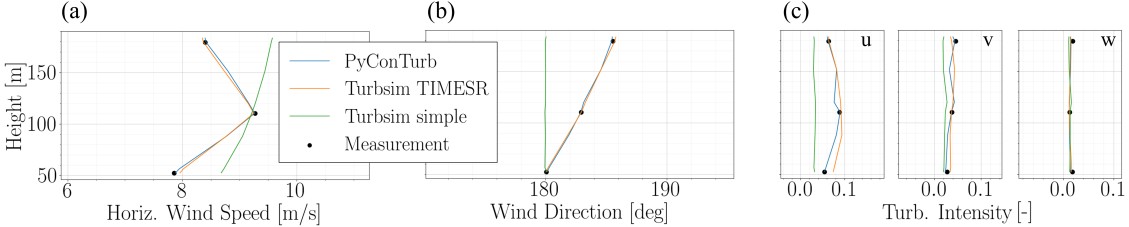

**Figure 9.** Analog to Fig. 8 except for a sample 10-minute bin that includes a non-monotonic shear profile.

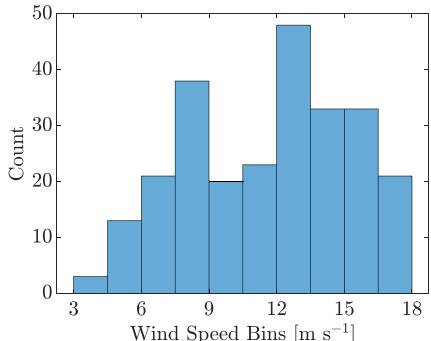

**Figure 10.** Histogram of the number of 10-minute bins within each of the wind speed intervals used for binning throughout Sect. 4

In the following subsections, the individual 10-minute bins have been sorted into wind speed intervals of 1.5 m s$^{-1}$ based on the mean horizontal wind speed of the ultrasonic anemometer near hub height. A histogram of the bin counts at each wind speed is shown in Figure 10.

### 4.1 Basic operability

First we considered the basic operability of the turbine models in terms of the controller set points for rotor speed and blade pitch. Figure 11 shows statistics of the model errors over the 253 10-minute bins compared to the measurements for rotor speed and blade pitch. For the rotor speed, the median errors within each bin are less than 0.5 rpm, or within 4 %. This error is not very sensitive to the inflow assimilation method, though the spread of error (i.e., the interquartile range) is smaller for the higher-fidelity assimilation methods. For the blade pitch, all the inflow assimilation methods underpredict pitch above rated, especially TurbSim simple, which again has a significantly larger spread than the other two methods. The cause for the underpredicted pitch by all three models in Region III (as well as the underpredicted rotor speed in Region II) could be related to aerodynamic modeling errors that produce an underestimation of the aerodynamic torque.

Figure 12 is analogous to Fig. 11 except it shows errors in the standard deviation rather than the mean. The median error in the standard deviation of rotor speed is less than 0.3 rpm, or 2 % of the nominal rotor speed at rated. The maximum error in the standard deviation of blade pitch is around 0.5°. Note that the underprediction of the standard deviation of blade pitch



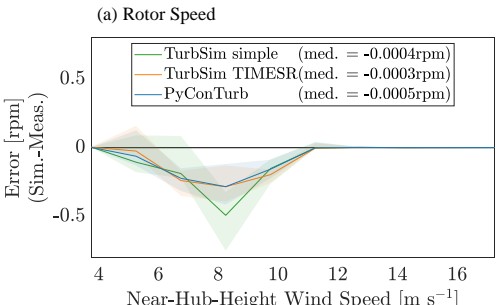 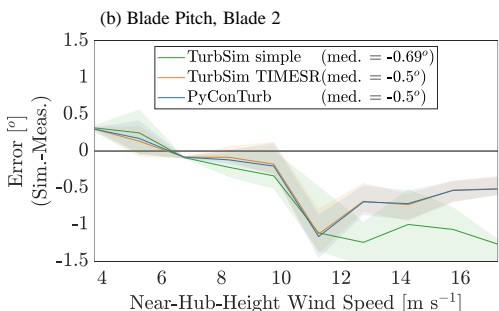

**Figure 11.** Comparison of (a) mean rotor speed and (b) mean blade pitch between models and measurement. The plots consist of statistics of each wind speed bin including the median (solid lines) and interquartile range (shaded areas). The median value reported in the legend is the overall median error across all bins.

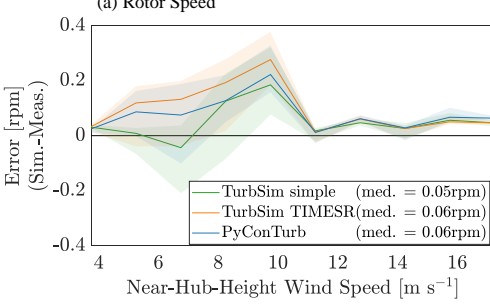 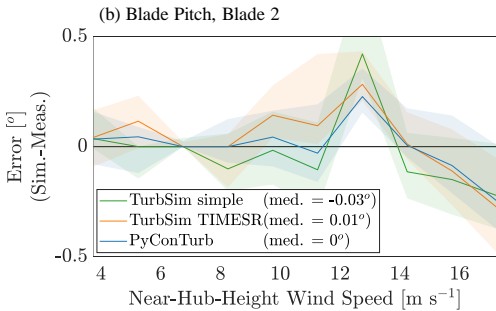

**Figure 12.** Comparison of (a) standard deviation of rotor speed and (b) standard deviation of blade pitch between models and measurement. See Fig. 11 for explanation of the lines, shading and legend.

at wind speeds $\geq 14$ m s$^{-1}$ appears to be a result of the omission of an IPC model in the controller, but the relatively small overall magnitude of the errors in this quantity corroborates the previous statement that few bins had persistent IPC activity in this data set.

## 4.2 Power

Figure 13 shows the comparison of simulated to measured power. The scatterplot in panel (a) indicates generally good performance of the models with some scatter between 6 and 12 m s$^{-1}$ that could be related to, for instance, spanwise inhomogeneity in the inflow that cannot be captured by the vertically aligned met tower sensors, or by the aforementioned temporal offset caused by the advection time of the flow between the meteorological tower and the rotor. In addition to the scatter, panel (b)





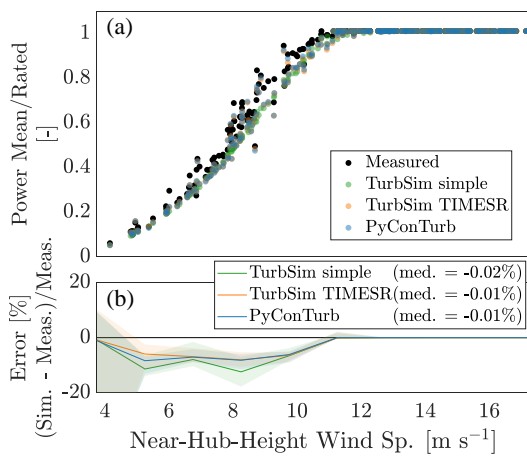

**Figure 13.** Comparison of (a) mean electrical power and (b) mean electrical power error between models and measurement. Each dot in (a) represents a 10-minute time series. See Fig. 11 for explanation of the lines, shading and legend in panel (b).

reveals a negative bias in the median modeling error in Region II that is up to 8.5 % for the higher-fidelity inflow assimilation methods and 12.5 % for the simpler method, and the shape of the error plots understandably bear some resemblance to those of the rotor speed errors in Fig. 11(a). The source of the bias is not presently known but could plausibly be related to errors in modeling of the controller, blade twist, airfoil performance, or rotor aerodynamics. Related to the latter, Madsen et al. (2012) noted an underprediction of power from BEMT formulations due to the lack of ground effect. We note also that the underprediction of the models at the knee of the power curve may be a result of exaggerated modeling of the peak-shaving strategy of the proprietary field controller.

### 4.3 Blade loading

Figure 14(a) shows the near-root flapwise moment mean at blade 1. The comparison shows that median modeling errors in each wind-speed bin are within 10 %, and the underprediction in Region II somewhat follows the underprediction of rotor speed and power in this region. The mean near-root edgewise moment is close to zero and is not reported.

The comparisons of the unsteady blade loading are shown in Fig. 14(b) and (c). The edgewise comparisons in panel (c) show little sensitivity to the inflow assimilation method, which is congruous with Rezaeiha et al. (2017), whose study demonstrated that aerodynamics (i.e., turbulence, wind shear, and yaw) account for <20 % of lifetime equivalent fatigue loads in the edgewise direction. Rather, it is rotor imbalances and gravity that dominate the edgewise fatigue budget. Thus, the <5 % simulation error for edgewise DEL suggests that the blade model development in Sect. 2.2 was successful in terms of edgewise characteristics. On the other hand, the flapwise fatigue comparison shown in Fig. 14(b) shows significant errors on the order of 5–18 % for the overall medians.



The authors believe that a significant amount of the overprediction of flapwise DELs stems from the computation of induction in OpenFAST. The last 10 years have seen awareness of overprediction of some unsteady QoIs from BEMT models versus higher-fidelity approaches, especially in sheared conditions (Madsen et al., 2012; Boorsma et al., 2016; Perez-Becker et al., 2020; Madsen et al., 2020). Although this overprediction seems to be improved by computing induction locally around the azimuth rather than using an annulus-averaged approach (Madsen et al., 2012, 2020), Perez-Becker et al. (2020) suggest that such locally-computed induction fields as found in OpenFAST, which include induced velocities from bound and wake vorticity, are still not accurate. They observed the 1P fluctuations of local angle of attack in OpenFAST to be overpredicted and found OpenFAST to consequently predict 9 % higher lifetime DELs for the out-of-plane blade root and the tower-base fore–aft bending moments compared to a lifting-line free-vortex method.

Possible evidence of this effect in our results is shown in Fig. 15, which re-plots the modeling error for flapwise DEL observed in Fig. 14(b) but this time as a function of turbulence intensity and wind shear exponent. The flapwise DEL errors show a positive correlation with wind shear exponent in Fig. 15(b), as expected from the above discussion. It is important to note that the omission of modeling of the field turbine's IPC actions could also produce this trend, but investigation by the authors on this data set suggests that IPC is not strongly active in most bins and does not account for most of the error observed in Fig. 14(b). The flapwise DEL error also has a weak negative correlation with turbulence intensity, as shown in Fig. 15(a), which could be related to the general decrease in the magnitude of wind shear, as well as the general decrease in relative significance of wind shear in the flapwise fatigue budget according to Rezaeiha et al. (2017), as turbulence increases.

Irrespective of the correlation between flapwise DEL error and wind shear in Fig. 15(b), there still exists significant sensitivity to the inflow assimilation method. The results from Rezaeiha et al. (2017) for a turbulence intensity similar to the mean of our study (i.e., ∼8 %), show that around half of the contribution to flapwise lifetime equivalent fatigue loads was from turbulence, and more than 8 % was from wind shear. Thus, the spread between inflow assimilation methods in flapwise DEL shown in Fig. 15 is not surprising since turbulence and wind shear are handled uniquely in each method.

Extrapolating the error trends in Fig. 15(b) to the point of zero wind shear suggests that the simplest inflow assimilation method (i.e., TurbSim simple) is validating better than the higher-fidelity methods in terms of flapwise DELs, and the apparent reason for the lower flapwise DEL predictions of TurbSim simple is the notably lower turbulence intensity of this approach stemming from the detrending process (see panel (c) of Figs. 8 and 9 and the first footnote of Table 3). Pedersen et al. (2019) found a similarly surprisingly result when comparing unsteady QoIs between simulations with unconstrained and constrained turbulence from a meteorological tower, and they concluded that measurements taken at large distances from the turbine should not be used to constrain turbulence because of possibly invalid assumptions related to frozen turbulence between the inflow measurement and turbine and those related to the measured flow field passing completely through the rotor disk. In our dataset, the results from Region II in Fig. 12(a) and Region III in Fig. 12(b) (before 14 m s$^{-1}$ when IPC actions in the field turbine increase) indicate an overprediction of unsteadiness in the turbine set points, despite the fact that the field turbine controller regulates the rotor speed more tightly than the modeled controller in low-wind-speed conditions. A hypothesis is that the flapwise DEL error could therefore be related to the BEMT formulation, which, in contrast to how it modifies the mean velocity with a rotor induction model, does not account for changes to the relative magnitude of the fluctuating component of



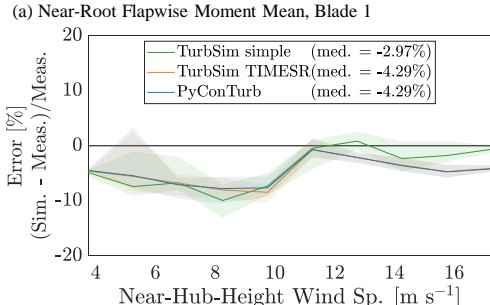 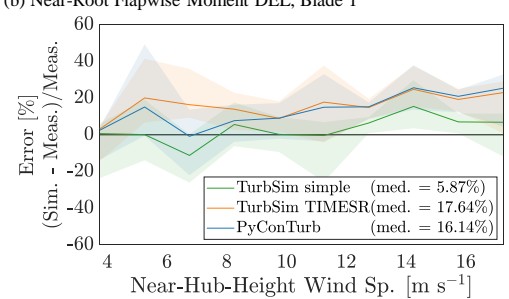

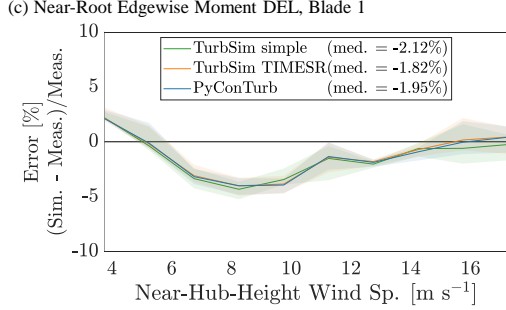

**Figure 14.** Comparison of (a)-(b) flapwise and (c) edgewise bending moment QoIs for blade 1 at 1.25 m from the root. See Fig. 11 for explanation of the lines, shading and legend.

the velocity (i.e., stretching of turbulence structures) due to the induction field of the rotor (see Mann et al. (2018) for relevant discussion).

Another modeling error that may contribute to the existence of the overprediction of flapwise DELs for the higher-fidelity inflow assimilation methods even at zero wind shear is the higher aeroelastic flapwise frequency of the blades as modeled than as measured as in Fig. 2. Higher aeroelastic frequency should increase fatigue damage similarly to how higher blade stiffness increases fatigue damage as noted in the early study by Noda and Flay (1999).

    The combination of modeling errors discussed in the last two paragraphs could be responsible for the $\sim$12 % residual
median error in the flapwise DELs at zero wind shear for the higher-fidelity inflow assimilation methods. Applying a similar offset to the error of the TurbSim simple results would then drop this method's flapwise DEL error to more than 12 % below zero, implying the existence of compensating errors in this method. Such errors could be related to TurbSim simple's inability to replicate non-monotonic shear profiles (see panel (a) of Fig. 9) and linear veer profiles (see panel (b) of Fig. 9) that would increase unsteady flapwise loading at the 2P frequency. Also, TurbSim simple's pre-defined spectral model leaves open the
possibility that the energy content at the lowest frequencies of the spectra is too low compared to a spectra derived from time-resolved measurements (see the comparison of the unconstrained Mann model to TIMESR in Nybø et al. (2021)).





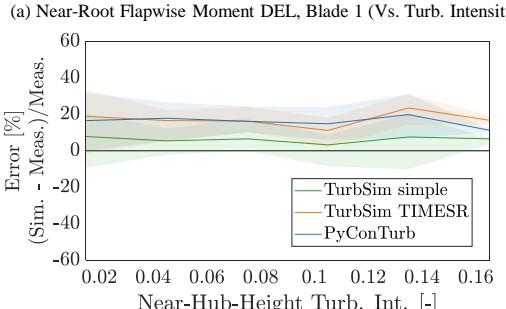 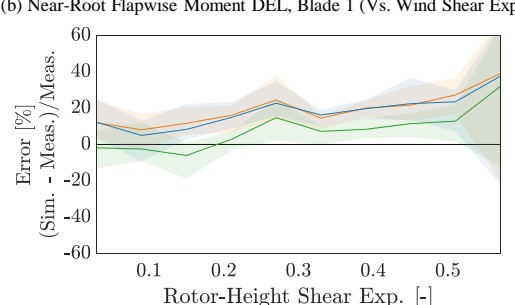

**Figure 15.** Comparison of the flapwise bending moment DEL from Fig. 14(b) except plotted versus (a) near-hub-height turbulence intensity and (b) rotor-height wind shear exponent. See Fig. 11 for explanation of the lines and shading.

## 4.4 Tower loading

Figure 16 shows the comparison of tower-base bending moments. The median errors of the model in panel (a) are generally less than ±10 % and indicate that the steady loading on the rotor and tower are well modeled, granted the underprediction of

steady tower loading in Region II that is related to the underprediction of rotor speed, power, and flapwise mean loading in this region as noted above. Note that the more negative mean error for the PyConTurb cases compared to the TurbSim ones is a result of the absence of tower nodes and aerodynamics in PyConTurb as described previously, and this effect grows with wind speed. The DEL errors of the model in panel (b) have a significant positive bias similar to that of the flapwise DEL errors. As before, the two higher-fidelity models show higher bias than TurbSim simple.

Similar hypotheses can be made for the overprediction of tower fore-aft DEL as for the overprediction of flapwise DELs in Sect. 4.3. The median errors for the fore-aft DELs in some wind-speed bins are even larger than those for the flapwise DEL, which could be related to exaggeration of the unsteadiness of the inflow by the BEMT model as suggested previously, especially since the tower-base fore-aft DELs are known to be highly sensitive to the accuracy of the wind spectrum (Nybø et al., 2021). Additionally, the aeroelastic fore-aft frequency in Fig. 2 is consistently overpredicted versus experimental values,

and this overprediction likely contributes to the overpredicted fore-aft DELs. The peak in DEL error at the rated power of the turbine could be a consequence of the absence of a tower damper model in the simulations.

## 5 Suggestions for a future experiment

Below are recorded lessons learned from the preceding analysis that may aid the design of future experiments.

### 5.1 Inflow measurements

This study leveraged meteorological tower data to define turbulence grids. Shortcomings of this approach include low spatial resolution and negligence of the effects of the rotor induction on the characteristics of the inflow fluctuations. An improved





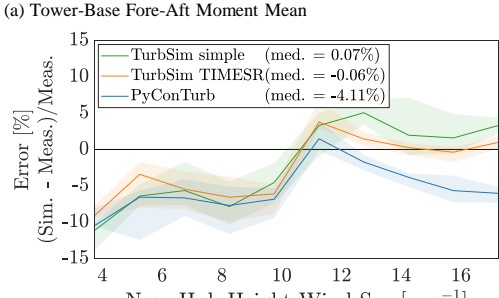
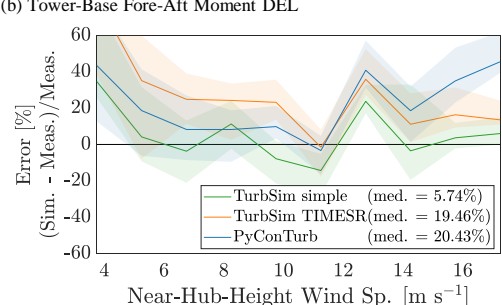

**Figure 16.** Comparison of fore-aft bending moment QoIs for the tower at 8.6 m from the base. See Fig. 11 for explanation of the lines, shading and legend.

experiment might include on-blade pressure probes as in Pedersen et al. (2019) to validate the induction physics predicted by the model. Taking a series of measurements as the flow moves toward the rotor such as with a nacelle- or hub-mounted lidar could additionally allow for step-by-step tracking of the inflow spectrum, shear, and veer, as well as comparison thereof with

the same quantities predicted by BEMT models.

Further, on-blade surface pressure measurements would be useful to quantify errors related to airfoil polars and three-dimensional effects. Tracking of the level of soiling on the turbine blades during the measurement period might also lead to more informed selection of the roughness condition of airfoil polars to be used in models. Better information about the local blade aerodynamic behavior will lead to improved estimates of aerodynamic frequencies and damping.

## 5.2 Experimental modal analysis

From the initial modal analysis discussed in Sect. 2.3.1, areas for improvement were identified. As mentioned, time-synchronous averaging was unsuccessfully used to isolate the blade damped natural frequencies from the rotational dynamics, which tend to dominate the spectral content. In the future, this method can be improved by increasing the data acquisition sampling rate for intervals of interest. Using an increased sampling rate will also allow for other time-domain methods to be leveraged. Another

method for future use will be order-domain analysis where the resulting time-series data can be analyzed on a per-revolution basis, which is directly related to the rotational speed of the rotor, allowing for separation of rotor speed harmonics from rotor structural frequencies. A final method for future consideration is operational modal analysis. This method can be used not only to understand the spectral content for structural frequencies but also to estimate the forced input into the rotor. A more detailed understanding of the modes can be gleaned where frequency, damping, and mode shape (where spatial resolution is adequate)

are estimated.

To develop a full-turbine modal estimate, the instrumentation effort should be focused on utilizing accelerometers. Postprocessing data from accelerometers installed along the blade span can be challenging, but their addition will allow for a modal map of acceptable spatial resolution to resolve mode shapes. Adding accelerometers to the hub, bed plate, and tower top will





also provide insight into the stiffness and damping of the coupled components. Accelerometers should also be installed before

and after the bearings, such as yaw and pitch, to quantify the impact of those degrees of freedom.

The installation of accelerometers is also not trivial. The blade, for example, provides limited entry, which makes it difficult to install sensors beyond 18 m for the turbine in consideration. Exterior installation allows for instrumenting the full span of the rotor but comes with installation, maintenance, and safety challenges for an operating turbine.

An improved method to understanding the full turbine system and component-damped natural frequencies is through an

in situ modal test. This test would be designed to excite the entire turbine with a known input force. Testing of this type creates a true frequency response function of the system where damping can more easily be approximated using common modal parameter estimation techniques. Also, modal shapes can be extracted, and modal scaling can be more accurately approximated.

Lastly, to achieve a full turbine excitation, a snap-back test method can be used where a reaction mass applies tension to

the turbine bedplate with a load cell in line with the tension. A quick-release device is used to release the mass and excite the turbine. In this way, an impulsive excitation of known force is applied to the turbine, and frequency response functions can be determined.

## 5.3 Controller modeling

The model controller used above was an open-source controller that was tuned to match the proprietary field controller. The

complexity of the proprietary controller is significant and resulted in simplifications and omissions in the model controller. Future work might benefit from using the DLL controller file from the field turbine, if permitted.

## 6 Conclusions

This study was designed to answer two questions: what is the value of one-to-one time-series-matched inflow for aeroservoelastic simulation, and what are the residual errors in turbine QoIs when modeling a 2.8 MW land-based wind turbine in

OpenFAST? The work included ingestion of 253 10-minute bins of operational data from a prototype wind turbine manufactured and operated by GE Vernova, as well as development of corresponding full-field turbulence grids from three levels of fidelity of inflow assimilation. The flow fields were input to an OpenFAST model that was developed to mimic as closely as possible the behavior of the field turbine. Subsequent bin-by-bin validation revealed that the median errors of steady QoIs for power, blade loading, and tower loading were generally within 5–10 %. The unsteady loading QoIs showed mixed results.

Simulated edgewise blade-root DELs were consistently predicted with less than 5 % error. However, the simulated flapwise blade-root DELs and tower-base fore-aft DELs showed a significant bias of 5–40 % overprediction, which the authors speculate could be a result of inaccurate aerodynamic modeling in sheared conditions (note this shortcoming is being addressed currently by NREL in ongoing development of OpenFAST), combined with possible errors in the simulated inflow wind fluctuations and aeroelastic flapwise and fore-aft frequencies. Interestingly, the lower-fidelity inflow assimilation technique produced the

lowest errors for the above two unsteady QoIs, which is similar to Pedersen et al. (2019). Since the higher-fidelity approaches

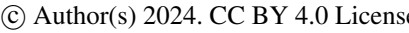

intuitively allow for more faithful representation of inflow featuring prominent coherent structures and/or non-stationarity, the possibility of the existence of compensating errors in the lower-fidelity approach should be considered. New approaches are under development to further investigate the origin of the errors in the unsteady loading QoIs and to determine the level of fidelity required by inflow models to accurately predict specific QoIs. Targeted measurements of inflow and blade quantities during the RAAW campaign are designed to narrow these modeling gaps. The result of this ongoing work will be a more physical approach to the aeroelastic simulations that are at the center of design and certification processes for wind turbines.

*Code and data availability.* The simulation codes used in this work are open-source and can be found at https://github.com/OpenFAST/openfast and https://github.com/NREL/ROSCO. OpenFAST v3.4.1 and ROSCO v2.7.0 were used in this study. The specific turbine model used is proprietary. The data generated in the field are also not publicly available.

*Author contributions.* KB processed the experimental results, analyzed the validation comparisons, and led the writing of the manuscript. PB, EB, and MC led the development of the OpenFAST model, conducted the verification and validation steps presented in Sect. 2, and helped write the manuscript. SD and CI led the experimental modal analysis. NdV ran the simulations and helped with the model validation. PD and NH lead the RAAW experimental campaign. JJ leads the development of OpenFAST. CK co-leads RAAW and supervised this validation study. DZ led the tuning of the ROSCO controller and helped with the model validation. All authors were a critical element of the research team and all contributed to the manuscript.

*Competing interests.* At least one of the (co-)authors is a member of the editorial board of Wind Energy Science.

*Acknowledgements.* This research was supported by the Wind Energy Technologies Office of the U.S. Department of Energy Office of Energy Efficiency and Renewable Energy. Sandia National Laboratories is a multimission laboratory managed and operated by National Technology & Engineering Solutions of Sandia, LLC, a wholly owned subsidiary of Honeywell International Inc., for the U.S. Department of Energy's National Nuclear Security Administration under contract DE-NA0003525. The views expressed in the article do not necessarily represent the views of the U.S. Department of Energy or the United States Government.

A portion of the research was performed using computational resources sponsored by the Department of Energy's Office of Energy Efficiency and Renewable Energy and located at the National Renewable Energy Laboratory. This work was authored in part by the National Renewable Energy Laboratory, operated by Alliance for Sustainable Energy, LLC, for the U.S. Department of Energy (DOE) under Contract No. DE-AC36-08GO28308. Funding provided by the U.S. Department of Energy Office of Energy Efficiency and Renewable Energy Wind Energy Technologies Office.

The views expressed in the article do not necessarily represent the views of the DOE or the U.S. Government. The U.S. Government retains and the publisher, by accepting the article for publication, acknowledges that the U.S. Government retains a nonexclusive, paid-up,



irrevocable, worldwide license to publish or reproduce the published form of this work, or allow others to do so, for U.S. Government
purposes. The technical and financial support of GE Vernova is gratefully acknowledged.



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
