# Peer review of "One-to-one aeroservoelastic validation of operational loads and performance of a 2.8 MW wind turbine model in OpenFAST"

_Wind Energy Science, 2023_

## Referee Comment (RC3)

Review of the paper: "One-to-one aeroservoelastic validation of operational loads and performance of a 2.8 MW wind turbine model in OpenFAST" by K. Brown et al.

Comments

The paper is extremely interesting and important for the wind energy community. It presents a comparison between the output of an aeroservoelastic simulator and the measurements taken on a 2.8MW wind turbine. The comparison is mainly focused on blade root and tower base loads and considers different methodologies to generate a full inflow field from a few point-wise measurements provided by a met-mast. The amount of analyzed data is significant, and this represents in my opinion the most important value of the present work.

The paper is well-written. Clearly, the model and the measurements cannot be shared and, hence, the results cannot be reproduced. Still the paper serves as a guideline in case other researchers need to perform a similar validation for different turbines.

I recommend accepting the manuscript with some minor modifications, which I listed here below.

Minor comments:

1. Pag. 4, section 2.1: Nothing is written concerning the aerodynamics of tower and nacelle. Please, write just a couple of sentences to if these are taken into account for completeness.
2. Figure 1: it is clear why the Author cannot display the y-axis values, but it could be important to provide numerical data related to the comparison. For example in Fig. 1, one can show the relative errors between prediction and measurements at least for the outer 60% of the blade, where displacements and rotations are significant.
3. Page 5-6, lines 134-136: Is it possible to provide the percentage increment of the blade structural damping that was necessary to use?
4. Page 5-6, lines 134-136: structural damping does not affect only the onset of instability, but, more in general, has an important impact on the vibratory level of the blades. Are the authors sure that the increase in the structural damping did not significantly impact the load estimation? From this point of view, one could use the structural damping as a tuning parameter that may adjust the load and even DEL prediction. Please, comment.
5. Page 7, line 147. How is the shaft and drive-train system modeled? Did you use beam-like or lumped elements? Moreover, why wasn't the torsion of the tower modeled? I assume that it was done because the tower is sufficiently rigid in torsion. If so, it should be written.
6. Pag. 7, line 157: "without the influence of the wind"-> isn't it more correct to say "in a standstill condition"? The wind probably is still blowing but the system does not rotate.
7. Pag 7, line 158: "The normal operation data were binned by rotor speed between cut-in and rated": it seems that full power region, between rated and cut-out was not considered.

This is unusual as load and DEL are typically significant in that operational region. Please, clarify.

8. Page 7, line165: "It was particularly difficult to find the first and second blade-root flap frequencies during normal operation", I think this is normal as blade flap (and in general out-of-plane modes, including whirling modes) is typically strongly damped by the aerodynamics, hence one does not see "the peak" in the spectrum. Moreover, how are the frequencies extracted from the data? Did you just peak-pick the mode from the PSD plot? If so, one should consider that the highest point in the PSD does not correspond to the frequency of the mode if the damping factor is high (>0.2). The maximum value of the PSD moves to a lower frequency as the damping factor increases. This may also be the reason for the poor matching observed for the out-of-plane frequencies.

9. Pag. 9. Section 2.4: Nothing is said about the dynamics of the actuators. Was this included?

10. Pag 9-10: section 3.1: Is it possible to extract from the comparison of cup and sonic anemometers some indication of the horizontal shear layer, that may have, if persistent, a huge impact on the periodic loads of the turbine?

11. Pag 17, line 340: Mismatches in the controller behavior are here excluded as possible sources of the errors between predicted data and measurements. Please, explain.

12. Section 4.3: fig. 14. While the mismatch in the mean value can be explained by the lower power, it is still difficult to understand the actual impact of the IPC on the DEL. Does it make sense to make an analysis removing the data affected by the IPC?

13. Section 5.2: In my opinion, also vibration data in standstill can be useful, to verify the modal content of the machine excluding the periodicity entailed by the rotation.

---

## Referee Comment (RC4)

[referee-annotated manuscript omitted]

---

## Author Comment (AC1)

**Color code:**

Black – referee comments

Green – authors' response to referee(s)

Blue – text added to the manuscript

Red – text deleted from the manuscript

**General comments:**

- For many of the observed differences potential explanations are given without substantiating, it would be good to make the paper less speculative. (RC1)

  The authors agree with this comment and have reduced the speculation surrounding supposed causes for the overprediction of flapwise DELs, speculation about the causes for the underprediction of power in Region II, as well as ill-advised assertions pinpointing hypotheses to numeric values without argumentation. Rather than describe each of these changes in detail in this General Comments section, we have described each one inline below where the reviewer made Specific Comments. Thank you for helping to improve this article.

- The figures on the results, from Fig 6 to the end, can be revised to make them easier to read (larger, different line styles, etc.) (RC1)

  Yes, the figures in the original manuscript were difficult to read, and they have now been resized and reformatted to be easier to read from Fig. 6 until the end.

- While the studies are relevant for the community, I do feel the paper is not straightforward and tends to bring in some other focus to the readers, especially at the beginning of the discussion. I believe the main objective and focus shall be the effects of different inflow conditions, and the authors may remove most parts of the paper which do not support the discussion. (RC2)
- In fact, extend the discussion of the turbine loads and response. Constraining the timeseries of the wind is an interesting approach which may be used to better understand the turbine behavior. (RC2)

  The authors have reworked the discussion of the turbine loads and its response to different inflow conditions. Specifically, in Section 4.3 the speculative discussion about the source of flapwise DEL errors is curtailed, and we have instead added

more quantitative discussion about why we do not believe the omission of IPC in our controller is primarily to blame for the overprediction of flapwise DELs:

*More specific insight on the effect of IPC on the flapwise errors in this dataset can be gained by an additional filtering step. One indicator of the strength of IPC activity is the standard deviation of the difference of two blades' pitch signals. Filtering out the half of the 10-minute bins with the larger such values and recomputing error metrics only reduced the range of median flapwise errors reported in Fig.~\ref{results_bladeloading_flapDELdetail}(a) from 8.57-17.46\% to 7.78-14.43\%. This result suggests that the omission of IPC may not sufficiently explain the errors in the flapwise DEL predictions.*

Further, we have added a new figure showing time series comparisons of flapwise loading between the measurement and three inflow assimilation methods. Taken together, we believe these changes make the discussion section (as well as the overall narrative) more straightforward and compelling. Please let us know if you do not agree.

- I think a similar story was/is under investigations within the IEA Wind Task 47 consortium. Probably providing some more link toward similar studies/report will be of benefit for the audience. (RC2)

  Indeed, IEA Wind Task 29 and 47 have discussed the topic of assimilating wind inflow measurements for use in wind turbine model validation. While related results have not yet been published on the work of Task 47 (benchmark simulations and validation from Task 47 are forthcoming), the main outcomes from Task 29 have already been cited in our Introduction, and include Schepers et al., 2021, Asmuth et al., 2022, and Boorsma et al., 2023. Of these works, the only one to use a constrained turbulence approach was Asmuth et al., 2022 (also summarized in Schepers et al. 2021), but the application in that work was quite different, focusing on the impact of an upstream wake on the blade loading of a downstream rotor.

- Last but not least, since the wind is constrained, I would expect some time domain comparison of the loads, not just the overall statistics (which can be achieved anyway without constraining in usual manner). (RC2)

  This comment is well taken, and we have added timeseries comparison of blade-root flapwise moment in Section 4.3. Please see Figure 18 in the new manuscript for this comparison.

- As mentioned in page 14, the advection time from the met tower to the turbine is completely omitted. It is claimed that this time ranges between 20-100s. However,

100s is not a negligible time interval. By doing so, all the benefits from assimilating the actual wind field are compromised. If there is a 100s shift between the simulated wind and the actual wind experienced by the turbine, then the analyzed loads and the resulting DELs correspond to totally different load time series which are only statistically equivalent as in the case of TurbSim simple. Could it be that the better agreement in this case by the assimilated inflow is because of the better representation of the spatial variation of the inflow? In the reviewer's opinion at least the frozen turbulence approach could have been considered in order to mitigate this effect. This is one key point that deserves further discussion. Perhaps by comparing not only the simulated wind speed time series but also some representative load series. (RC4)

This is a good point, and we have addressed by recalculating the results to account for the advection times as described in the new paragraph in Section 3.3 and copied below:

*A temporal offset was then applied to the simulation timeseries to account for the advection time of the flow between the meteorological tower and the rotor by leveraging knowledge of the location of the meteorological tower relative to the turbine and by assuming the flow moved with the 10-minute mean measured wind direction and speed at the near-hub-height sonic anemometer on the meteorological tower. Both the experimental and simulation results were then shortened by 20--100 s depending on the wind speed so that only the overlapping segments of time between the simulated and measured channels were retained. Although the above process is a simple temporal correction with some inaccuracies, we expect any residual errors in the temporal alignment to add noise but not bias the sampling of a large number (i.e., 253) of 10-minute bins drawn from all times of day.*

Further, we have added a timeseries comparison of turbine blade-root flapwise moment in Section 4.3. Please see Figure 18 in the new manuscript for this comparison, which demonstrates the ability of the constrained turbulence methods to track the long-pass temporal variations in the inflow.

- The authors claim that the overprediction in the flapwise moment DELs is due the induction modelling in openFAST which fails to correctly predict azimuthal variation of the induction at high shear exponent values. However, even at low shear exponent values the overprediction is substantial (~20%). Another point is that overprediction is equally high both at low and high wind speeds where induction is overall much lower (normally its variations will be lower too). An easy explanation could be that the overprediction is due to the omission of IPC. This viewpoint is refuted thought by the fact that deviation is independent of turbulence intensity. Finally, in the reviewer's opinion this point remains not properly answered while some of the possible reasons claimed by the authors are not supported by results

(e.g. not matching flapwise frequency). I would recommend some further elaboration on that matter or maybe some more concrete explanation (perhaps leveraging TurbSim simple results which seems to do better). (RC4)

The reviewer's point is well taken. While we do not have a bona fide explanation for why the flapwise DELs are overpredicted, we have added some more concrete explanation as to why we do NOT think the main problem is the omission of IPC in our controller. The following added excerpt explains this line of thought:

*The omission of modeling of the field turbine's IPC actions could produce this trend since IPC has been shown in some cases to reduce flapwise DELs on the order 10-20\% \citep{bossanyi2003individual,van2005individual}. Indeed, the flapwise DEL errors fall closest to zero for all three inflow assimilation methods as the inflow shear exponent (and thus the likelihood of IPC activity) tends to zero. More specific insight on the effect of IPC on the flapwise errors in this dataset can be gained by an additional filtering step. One indicator of the strength of IPC activity is the standard deviation of the difference of two blades' pitch signals. Filtering out the half of the 10-minute bins with the larger such values and recomputing error metrics only reduced the range of median flapwise errors reported in Fig.~\ref{results_bladeloading_flapDELdetail}(a) from 8.57-17.46\% to 7.78-14.43\%. This result suggests that the omission of IPC may not sufficiently explain the errors in the flapwise DEL predictions.*

We have also improved the surrounding discussion about other possible contributions to the flapwise DEL error, as well as adding plots of timeseries data of the flapwise loads, for reference. We believe these additions make the paper significantly stronger.

**Specific comments:**

*Section 1*

- Line 69 - In section 2 it is mentioned that proprietary controller is replaced by a general purpose controller (RC4)

  Yes, this was a misleading sentence. We modified it to be more truthful:

  *For the turbine aeroelastic and controller modeling, a significant amount of attention was devoted to matching the behavior of the field turbine by  incorporating proprietary information about the characteristics of the turbine and controller as closely as possible into the sub-modules of OpenFAST.*

*Section 2*

- -p3 section 2 p77/78 A description of the test set-up of the experiment including turbine, instrumentation and visuals would help the reader. What is the rated wind speed? Part of section 3.1 could be moved here. (RC1)

  Yes, a graphic showing the relevant instrumentation has been added as the new Figure 4. We have elected to put this figure in Section 3 because our goal was to wait until Section 3 to introduce the main experimental campaign. We have instead added a line in Section 2 to point the reader ahead to Section 3 for full details on the turbine setup.

[Figure]

*Section 2.1*

- Line 89 - I suggest to delete performance (RC4)
  This change has been implemented.

- In relation to the later observed differences and their hypothesized origin it would be good to add some more details (RC1)
  - Details of the airfoil data weighting process (% clean/rough?)

- o Some info on BEM implementation, e.g. considering local / annulus avg momentum

Section 2.1 provides an overview and relevant references of the dynamic blade element momentum theory used in OpenFAST. The blending of clean and rough polars was done internally at GE and we do not have and cannot share its details.

- Clarify what is meant by 'tuning' environmental conditions and its relevance for the aerodynamic modeling section. (RC1)

"Tuning" was meant to indicate that the inflow conditions such as air density were set for each bin according to measured values. We have changed the sentence as follows:

*The environmental conditions (i.e., inflow density and velocity) were set on a bin-by-bin basis using the data from the met tower (see Sect. 3)*

- Pag. 4 , section 2.1: Nothing is written concerning the aerodynamics of tower and nacelle. Please, write just a couple of sentences to if these are taken into account for completeness. (RC3)

Yes, there was some detail about the tower modeling omitted from the original manuscript. The is now text describing the details as copied below:

*The effect of the tower on the incoming wind was accounted for using OpenFAST's baseline potential flow model for two of the three inflow assimilation methods as discussed further below, and tower aerodynamic loading was calculated for these two methods, as well. The downstream tower shadow was not included in the modeling.*

The discussion further below that is referenced mentions:

*The TurbSim methods include tower nodes to simulate wind loading on the entire tower including the region below the turbulence plane \citep{jonkman2014turbsim}, whereas PyConTurb does not have this feature. Tower aerodynamics were therefore disabled for the PyConTurb cases.*

*Section 2.2*
- -p4/5 section 2.2.1 and 2.2.2 - This comparison should seriously be reconsidered in my opinion. What is the value of comparing against another dataset if no detail is given about it? (RC1)

We politely disagree with the reviewer. We think that Sections 2.2.1 and 2.2.2 describe relevant verification and validation studies. The goal of the verification study is whether our OpenFAST model matches against the model from GE. Both models are certainly not a true representation of reality, but having them match was a key milestone of our work and we believe that the results of that step should be reported for completeness.

-Numerical values from a GE tool without knowing its background
-A lab test without any info on the set-up

Unfortunately, we do not have and cannot share the details of the numerical tools or lab tests at GE. Nonetheless, these are the tools that were used to design and certify the wind turbine. We understand that this lack of details hinders the replicability of our work, but this is a limitation that was imposed (and is commonly imposed), and that we had to accept.

-Furthermore it is not clear what kind of load is under consideration here?

The elastic deformation of the blade described in Section 2.2.1 and in Figure 1 is subjected to gravity loading. This is explained at the beginning of section 2.2.1 as well as in the title and in the legend of Figure 1.

- -p4 line 115 - Can a twist difference of 0.5deg be considered small? (RC1)

  The solid and dashed lines from the bottom plots of Figure 1 overlap almost exactly. The difference of 0.5 deg is seen in the flapwise rotation (grey lines) at blade tip, where small differences in the meshing likely caused OpenFAST and the GE tool to diverge slightly. Note that we have changed the word "offset" with "difference", since the word "offset" points to a more systematic error.

- -p5 Fig.1 - If this plot still makes sense (see comment above), without values on the axis, also a plot highlighting the % difference would help to aid interpretation. (RC1)

  We understand the point of the reviewer that this work cannot be replicated by anyone in the community. Unfortunately, this is a fact that we have to live with and that we cannot resolve. We opted for an absolute plot with masked y-axis labels to show that the verification study returned a good match across the span of the blade. A plot with relative differences would be harder to interpret and would still not lead to any replicability. Overall, we think that this choice is a matter of taste more than a requirement. Please let us know if you disagree with our assessment.

- Figure 1 : it is clear why the Author cannot display the y axis values , but it could be important to provide numerical data related to the comparison. For example in Fig.

1 , one can show the relative errors between prediction and measurements at least for the outer 60% of the blade , where displacements and rotations are significant. (RC3)

We understand the point of the reviewer that this work cannot be replicated by anyone in the community. Unfortunately, this is a fact that we have to live with and that we cannot resolve. We opted for an absolute plot with masked y-axis labels to show that the verification study returned a good match across the span of the blade. A plot with relative differences would be harder to interpret and would still not lead to any replicability. Overall, we think that this choice is a matter of taste more than a requirement. Please let us know if you disagree with our assessment.

- -p5 line 134 - What was the originally specified damping and how was this modified? (RC1)
- Page 5 6 , lines 134 136 : Is it possible to provide the percentage increment of the blade structural damping that was necessary to use (RC3)
- Page 5 6, lines 134 136: structural damping does not affect only the onset of instability, but, more in general, has an important impact on the vibratory level of the blades. Are the authors sure that the increase in the structural damping did not significantly impact the load estimation. From this point of view, one could use the structural damping as a tuning parameter that may adjust the load and even DEL prediction. Please, comment. (RC3)

These are all valid concerns of the reviewers. We cannot share the initial absolute value provided by GE. In OpenFAST, the damping value in the edgewise direction was increased by almost an order of magnitude. This increase is certainly concerning, and it likely covers some discrepancies in the unsteady aerodynamic models. These discrepancies will be investigated in future work. The text in the article has been adjusted to reflect these thoughts.

*Section 2.3*
- Page 7, line 147. How is the shaft and drive train system modeled? Did you use beam like or lumped elements? Moreover, why was not the torsion of the tower modeled ? I assume that it was done because the tower is sufficiently rigid in torsion. If so, it should be written. (RC3)

Thank you for the valuable comment. The drivetrain was modeled as lumped masses, with the addition of a torsional degree of freedom. Tower torsion was not included because it is currently not available in ElastoDyn. We added two paragraphs in section 2.2.2 discussing these limitations.

- Pag. 7, line 157: "without the influence of the wind"--> isn't it more correct to say "in a standstill condition"? The wind probably is still blowing but the system does not rotate. (RC3)

  Great point, thank you for spotting this incorrect sentence. We have changed it to "minimizing the impact of rotor aerodynamics".

- Pag 7 , line 158: "The normal operation data were binned by rotor speed between cut in and rated": it seems that full power region , between rated and cutout was not considered. This is unusual as load and DEL are typically significant in that operational region. Please, clarify. (RC3)

  This binning refers to the experimental modal analysis, not to DEL computation. The analysis was limited to this range of wind speeds to conduct the comparison between experimental and numerical natural frequencies as a function of rotor speed, which only varies up to rated. We added a sentence about that.

- Page 7, line165: "It was particularly difficult to find the first and second blade-root flap frequencies during normal operation", I think this is normal as blade flap (and in general out-of-plane modes, including whirling modes) is typically strongly damped by the aerodynamics, hence one does not see "the peak" in the spectrum. Moreover, how are the frequencies extracted from the data? Did you just peak-pick the mode from the PSD plot? If so, one should consider that the highest point in the PSD does not correspond to the frequency of the mode if the damping factor is high (>0.2). The maximum value of the PSD moves to a lower frequency as the damping factor increases. This may also be the reason for the poor matching observed for the out-of-plane frequencies. (RC3)

  The authors agree with the reviewer regarding this comment. To extract the frequencies, we first identified the rotor passing frequencies and then used the design frequencies to aid us in the general range to focus on within the PSD plot. Even then, the modes were sometimes so damped that it was difficult to identify a distinguishable peak. The comment from the reviewer has been incorporated into the text to help provide more context why it was difficult to locate the peaks.

- -p7 line 169  - What is the sample frequency and what frequency would be needed? (RC1)

  The sampling frequency was 50Hz. For the TSA method, it would have been nice to have something in the order of 500Hz or more in order to provide enough resolution to apply the technique successfully.

- -p8 line 175  - Not sure which linearized unsteady aero model is referred to from the description in section 2.1? (RC1)

  We've added a reference to section 2.1 and to the relevant reference Branlard et al., 2022

*Section 2.4*

- Lines 199-201 - This is a bit surprising because the foreseen reduction in blade loads due to IPC is usually higher than 10-15%. Maybe some further explanation on how the IPC controller is working on this particular turbine to justify that the omission will not play any important role. (RC4)

  We understand the surprise, and we agree that past studies show reductions in flapwise blade DEL on the order of 10-15% with IPC. Justification for this omission has now been added further down in the article, and we believe this addition addresses the reviewers concern. In the context of whether the omissions of IPC is significant to the prediction errors for the flapwise DEL, we have added:

  *The omission of modeling of the field turbine's IPC actions could produce this trend since IPC has been shown in some cases to reduce flapwise DELs on the order 10-20\% \citep{bossanyi2003individual,van2005individual}. Indeed, the flapwise DEL errors fall closest to zero for all three inflow assimilation methods as the inflow shear exponent (and thus the likelihood of IPC activity) tends to zero. More specific insight on the effect of IPC on the flapwise errors in this dataset can be gained by an additional filtering step. One indicator of the strength of IPC activity is the standard deviation of the difference of two blades' pitch signals. Filtering out the half of the 10-minute bins with the larger such values and recomputing error metrics only reduced the range of median flapwise errors reported in Fig.~\ref{results_bladeloading_flapDELdetail}(a) from 8.57-17.46\% to 7.78-14.43\%. This result suggests that the omission of IPC may not sufficiently explain the errors in the flapwise DEL predictions.*

- -p9 line 202  - Clarify what is meant by 'more sophisticated system monitoring and control mode switching' (RC1)

  Because of unknown trade secrets about the GE controller, we cannot disclose the specific behavior witnessed in the SCADA data.  After reviewing specific cases, we can conclude that the ROSCO controller behaves similarly to the GE controller at low wind speeds, but the specific control actions are different, which can lead to different transient behaviors.  For clarity, we have removed these sentences and used clearer statements about the differences.

- Pag. 9. Section 2.4: Nothing is said about the dynamics of the actuators. Was this included? (RC3)

  Actuator dynamics are not included in the OpenFAST or ROSCO models. Instead, we tuned the pitch response directly to match the SCADA data.  Including the pitch actuator would have added an additional tuning parameter and could have produced a slightly better match, but the overall goal would have been the same.

*Section 3*
- Line 222 - What about nacelle position? (RC4)

  This is a good question. We did not process nacelle accelerometer data from the measurement campaign, and the authors agree that it could add to the narrative but have not included it at this point.

- Pag 9-10: section3.1: Is it possible to extract from the comparison of cup and sonic anemometers some indication of the horizontal shear layer, that may have, if persistent, a huge impact on the periodic loads of the turbine? (RC3)

  This is an insightful point. We did not consider using the lateral separation between the cup and sonic anemometers to assess the presence of time-local mean horizontal shear, though the authors agree with the suggestion. In a future study, this idea could be extended even further by incorporating additional constraint points in the PyConTurb simulations to represent all the available anemometers at the proper lateral separation. We considered this for the present study, but lack of vertical velocity information from the cup anemometers was a deterrent.

- -p11 line 260 - Quantify what is meant by some vs persistent IPC activity (RC1)

  We cannot clarify how the GE controller works on this turbine due to trade secrets. However, some justification for the omission of IPC in our controller is now provided further below in the manuscript, and we believe this addresses the reviewer's concern:

  *The omission of modeling of the field turbine's IPC actions could produce this trend since IPC has been shown in some cases to reduce flapwise DELs on the order 10-20\% \citep{bossanyi2003individual,van2005individual}. Indeed, the flapwise DEL errors fall closest to zero for all three inflow assimilation methods as the inflow shear exponent (and thus the likelihood of IPC activity) tends to zero. More specific insight on the effect of IPC on the flapwise errors in this dataset can be gained by an additional filtering step. One indicator of the strength of IPC activity is the standard deviation of the difference of two blades' pitch signals. Filtering out the half of the 10-minute bins with the larger such*

*values and recomputing error metrics only reduced the range of median flapwise errors reported in Fig.~\ref{results_bladeloading_flapDELdetail}(a) from 8.57-17.46\% to 7.78-14.43\%. This result suggests that the omission of IPC may not sufficiently explain the errors in the flapwise DEL predictions.*

- -p13 line 294 - Not sure what is meant here, if a wind field turbulent box is created, the appropriate tower incident wind velocity could be interpolated from that I suppose? (RC1)

  The problem is that the wind field turbulent box does not extend all the way to the ground while the tower does. So, without tower nodes as TurbSim offers, some of the wind loading on the tower at lower heights on the tower will not be taken into account, though we know that wind loading on the tower is small relative wind loading on the rotor. We have modified the text as follows to clarify this point:

  *The TurbSim methods include tower nodes to simulate wind loading on the entire tower including the region below the turbulence plane (Jonkman, 2014),....*

- Line 319-320 - This is up to 1/6 of the time series. In my opinion it substantially compromises any effort you have put in matching the time series through assimilation. Why not applying at least a frozen turbulence approximation? (RC4)

  The authors agree; please see our response to the major comment on this issue.

*Section 4*
- Fig 7 - Explain units (normalization applied) on y-axis (RC4)

  Yes, thank you mentioning this omission. The figure caption has been clarified to include the method used to calculate the y-axis.

- Pag 17, line 340: Mismatches in the controller behavior are here excluded as possible sources of the errors between predicted data and measurements. Please, explain. (RC3)

  Since in Region III operation the controller will hold rotor speed constant while adjusting its collective pitch until the rated torque is returned, a mean difference in pitch as we observed seems to indicate that the simulation is mis-predicting torque, and this should stem from an aerodynamic inaccuracy in the simulation. Please let us know if you disagree.

- -p17 line 342 and throughout manuscript - Not sure what is meant by Region II and III? (RC1)

This point has been clarified with an explanation at the beginning of Section 4:

*As the rated wind speed is ~11 m/s, there are roughly an even number of bins corresponding to above-rated wind speeds (i.e., Region III) as there are corresponding to below-rated conditions (i.e., Region II).*

- Line 352 "with some scatter between 6 and 12 m/s" - This is more or less the whole partial load range! (RC4)

  Yes, that is very true. The authors' intent here was to highlight that there is some variance in the predictions in this specific wind speed range rather than to suggest that the results in this range were of overall poor quality. We have modified the wording in an attempt to convey this:

  *The scatterplot in panel (a) indicates generally good performance of the models with increased scatter apparent below 10 m s$^{-1}$ that could be related to, for instance,...*

- Line 357 – "of the rotor speed errors in Fig 11a" - I cannot understand such resemblance unless the controller fails to track Cpmax. A difference in the rotor speed in the partial load region indicates a difference between actual and predicted TSR_opt. However, a similar difference in the power output indicates an underprediction in Cpmax which is a different story. Could you please explain this point? (RC4)

  This comment is insightful, and the authors agree. Given that there is no rationale for the purported resemblance of the rotor speed and power curve errors, and that we do not yet have a substantiated argument for why Cpmax is underpredicted in this region, we have removed this comment.

- -p19 Fig. 13 - A plot of power coefficient Cp would allow to zoom in better on differences irrespective of wind speed (RC1)

  We wholeheartedly agree that a Cp plot would be useful here. However, regretfully, our industry partner only gave permission to plot certain QoIs, and Cp is not allowed for this paper.

- -p19 line 359 - As mentioned, power differences can arise from a large number of reasons. Is there a particular reason to suspect the ground effect here, e.g. looking at past comparisons to field data? Acknowledging the corresponding trend differences in rpm (fig 11, below rated, ) and pitch (fig. 11 above rated), would that help to substantiate further? (RC1)

The authors acknowledge that singling out the ground effect is maybe not appropriate in this case. We have removed this comment altogether and instead leave open to possibility for errors due to:

*modeling of the controller, blade twist, airfoil performance, or rotor aerodynamics*

- Line 360 "of the peak-shaving strategy" – this makes sense (RC4)

  Thank you.

- Line 365 – "The mean near-root edgewise moment is close to zero and is not reported" - You should be able to identify differences similar to those reported for power. (RC4)

  This is a valid point. We do not report averaged edgewise mean loading because mean edgewise is hard to measure given the large fluctuations around a mean of low magnitude. Azimuthally-averaged mean edgewise loading might be a useful diagnostic in some situations though as the author suggests.

- Line 366 – "of the unsteady blade loads" - please directly refer to DELs (RC4)

  OK, the sentence now reads:

  *The comparisons of the* *near-root DELs are shown in…*

- Line 367 – "whose study demonstrated that aerodynamics" - However the increased damping considered is clearly reflected. This should be reported herein. (RC4)

  This is a good point, and we added a reminder about the artificial increase in edgewise structural damping.

- -p20 1st paragraph, overprediction of flapwise fatigue loads - It is hypothesized that differences arise from the BEM model in OpenFAST, in agreement with previous literature. Would it be worthwhile to compare against results from a different model like vortex wake for a particular case under investigation to substantiate this further? (RC1)

  The authors fully agree that a comparison with a higher-fidelity code such as a free-vortex wake code would help substantiate this claim. Unfortunately, the project did not allow this comparison to be made, but we believe it would be valuable for future

work. We have added a note at the end of the discussion about the possibility to use a FVWM in future work to substantiate the hypothesis:

*Further insight on this possible modeling discrepancy might be afforded by a higher-fidelity modeling technique such as a free-vortex wake method.*

- Line 373 – "The authors believe that a significant amount of the overprediction of flapwise DELs stems from the computation of induction in OpenFAST" – I don't quite agree because induction decreases with the wind speed (variations as well - if not explain) and therefore its effect on fatigue predictions should be decreasing as well. What about not including IPC? (RC4)

  This section of the manuscript has been reworked significantly. First, we have acknowledged and explained in more detail why we do not consider the omission of IPC to be the primary cause of the discrepancy of flapwise DELs, and this is explained in our response to your major comment above on the same topic. We understand your comment that the effect of induction on fatigue predictions should be decreasing moving further into Region III. We do not have a sound explanation as to why the errors in flapwise DEL persist and even increase at higher wind speed. It is possible that there are confounding effects present, or that the sample size per wind-speed bin is not large enough. To reflect the lack of understanding on our part, we have softened the language quoted by the reviewer above to instead read:

  *Another possible explanation is that overprediction of flapwise DELs stems from the computation of induction in OpenFAST.*

  Further, a line at the end of the paragraph now spells out the point the reviewer mentioned:

  *One aspect of the flapwise error that is not explained by the above induction argument is that the errors in flapwise DEL persist and even increase at higher wind speed where induction begins to decrease.*

- -p20/21 line 407/408 - Consider to reformulate 'changes to the relative magnitude of the fluctuating component of the velocity (i.e. stretching of turbulence structures) due to the induction field' for clarity. To my understanding the publication of Branlard (https://doi.org/10.1016/j.jweia.2016.01.002.) seems in contradiction to this observation, suggest clarification. (RC1)

  Thank you for the helpful comment; you are correct that our original assertion is not completely congruous with Branlard et al. (2016), who notes that there is a slight decrease of energy in smaller scales but not a significant overall change in the inflow spectrum due to the presence of the turbine. Further Mann et al (2018)

weighs in on this topic with measurement results, and the rotor is seen to have the effect of increasing the fluctuating component of streamwise velocity below rated while decreasing it above rated. In light of all this, we have rephrased this comment (as well as one further on relating to the tower QoI). It now says:

*A hypothesis is that the flapwise DEL error could therefore be related to the BEMT formulation, which, in contrast to how it modifies the mean velocity with a rotor induction model, does not account for changes to the relative magnitude of the fluctuating component of the velocity (i.e., distortion of turbulence structures) due to the induction field of the rotor. \citet{branlard2016impact} and \citet{mann2018does} consider this question, and their conclusions about the degree of amplification or attenuation of the turbulence spectrum due to the presence of the rotor varying depending on the turbulent lengthscale and the region of rotor operation (i.e., the slope of thrust coefficient curve versus wind speed). Further insight on this possible modeling discrepancy might be afforded by a higher-fidelity modeling technique such as a free-vortex wake method.*

- -p21 line 412 - Would the proximity of blade frequencies to excitation frequencies be important to consider as well? (RC1)

  Good question. The differences are observed mostly in flap quantities, where aeroelastic damping is high. So, we think this is not as important.

- Section 4.3: fig.14. While the mismatch in the mean value can be explained by the lower power, it is still difficult to understand the actual impact of the IPC on the DEL. Does it make sense to make an analysis removing the data affected by the IPC? (RC3)

  We agree with this suggestion and have added results of a mini study to provide some insight into the effect of omitting IPC. The added excerpt is as follows:

  *The omission of modeling of the field turbine's IPC actions could produce this trend since IPC has been shown in some cases to reduce flapwise DELs on the order 10-20\% \citep{bossanyi2003individual,van2005individual}. Indeed, the flapwise DEL errors fall closest to zero for all three inflow assimilation methods as the inflow shear exponent (and thus the likelihood of IPC activity) tends to zero. More specific insight on the effect of IPC on the flapwise errors in this dataset can be gained by an additional filtering step. One indicator of the strength of IPC activity is the standard deviation of the difference of two blades' pitch signals. Filtering out the half of the 10-minute bins with the larger such values and recomputing error metrics only reduced the range of median flapwise errors reported in Fig.~\ref{results_bladeloading_flapDELdetail}(a) from 8.57-17.46\% to 7.78-14.43\%. This result suggests that the omission of IPC may not sufficiently explain the errors in the flapwise DEL predictions.*

- -p21 line 414 - Hypothesizing about the cause for differences is commonplace in a scientific article. Suggested to be careful with pinpointing these to a numeric value without argumentation. (RC1)

  This advice is well-received, and we have reworded this to be a more general comment without a numerical value attached since we do not have extensive evidence.

- -p22 Fig. 15 Acknowledging the atmospheric relation between TI and shear, can we exclude bias here? Can we plot the trend against shear for a constant TI and vice-versa for a constant shear against TI. (RC1)

  This is a good comment, and the authors agree with the idea. This approach has not been implemented however due to the already low sample size per wind speed bin that makes the results already somewhat noisy. Rather, we hope that by plotting the flapwise errors vs both turbulence intensity as well as wind shear, readers will have a better if not complete understanding.

*Section 5*
- Section 5.2: In my opinion, also vibration data in standstill can be useful, to verify the modal content of the machine excluding the periodicity entailed by the rotation. (RC3)

  As mentioned in section 2.3.1, the emergency stop data that was collected was meant to help find the natural frequencies by exciting the modes with minimal effect of the rotor rotation.

- -p23 line 446-449 - Should the recommendation for surface pressure measurements be part of a section dedicated to inflow measurements? (RC1)

  It is agreed that this would be better under a different section, so we have moved that bit of text about surface pressure measurements to its own subsection titled: *On-blade measurements*

- -p24 section 5.3 - Control modeling should not be part of a section about Suggestions for a future experiment.  (RC1)

  OK, we have modified the section heading to be:

  *Suggestions for a future  validation campaign*